# CAUSAL FAIRNESS UNDER UNOBSERVED CONFOUNDING: A NEURAL SENSITIVITY FRAMEWORK

**Maresa Schröder, Dennis Frauen & Stefan Feuerriegel**
Munich Center for Machine Learning
LMU Munich
{maresa.schroeder,frauen,feuerriegel}@lmu.de

## ABSTRACT

Fairness of machine learning predictions is widely required in practice for legal, ethical, and societal reasons. Existing work typically focuses on settings without unobserved confounding, even though unobserved confounding can lead to severe violations of causal fairness and, thus, unfair predictions. In this work, we analyze the sensitivity of causal fairness to unobserved confounding. Our contributions are three-fold. First, we derive bounds for causal fairness metrics under different sources of unobserved confounding. This enables practitioners to examine the sensitivity of their machine learning models to unobserved confounding in fairness-critical applications. Second, we propose a novel neural framework for learning fair predictions, which allows us to offer worst-case guarantees of the extent to which causal fairness can be violated due to unobserved confounding. Third, we demonstrate the effectiveness of our framework in a series of experiments, including a real-world case study about predicting prison sentences. To the best of our knowledge, ours is the first work to study causal fairness under unobserved confounding. To this end, our work is of direct practical value as a refutation strategy to ensure the fairness of predictions in high-stakes applications.

## 1 INTRODUCTION

Fairness of machine learning predictions is crucial to prevent harm to individuals and society. For this reason, fairness of machine learning predictions is widely required by legal frameworks (Barocas & Selbst, 2016; De-Arteaga et al., 2022; Dolata et al., 2022; Feuerriegel et al., 2020; Frauen et al., 2023a; Kleinberg et al., 2019). Notable examples where fairness is commonly mandated by law are predictions in credit lending and law enforcement.

A prominent fairness notion is *causal fairness* (e.g., Kilbertus et al., 2017; Plecko & Bareinboim, 2022; Zhang & Bareinboim, 2018b). Causal fairness leverages causal theory to ensure that a given sensitive attribute does not affect prediction outcomes. For example, in causal fairness, a prison sentence may vary by the type of offense but should not be affected by the defendant's race. Causal fairness has several advantages in practice as it directly relates to legal terminology in that outcomes must be independent of sensitive attributes (Barocas & Selbst, 2016).[1]

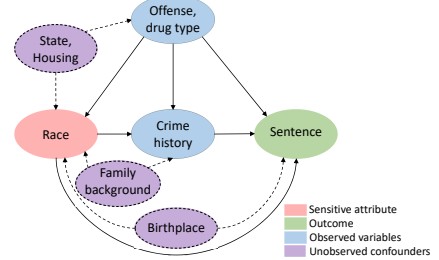

Figure 1: Example: Causal graph for predicting prison sentences.

Existing works on causally fair predictions (e.g., Nabi & Shpitser, 2018; Zhang & Bareinboim, 2018a;b) commonly focus on settings with *no unobserved confounding* and thus assume that the causal structure of the problem is *fully* known. However, the assumption of no unobserved confounding is fairly strong and unrealistic in prac-

---

[1] In the literature, there is some ambiguity in the terminology around causal fairness. By referring to our fairness notion as "causal fairness", we are consistent with previously established literature (Plecko & Bareinboim, 2022). However, we acknowledge that there are other fairness notions based on causal theory (e.g., counterfactual fairness), which do *not* present the focus here. More details about the difference are in Supplement C.1.

tice (Carey & Wu, 2022; Fawkes et al., 2022; Kilbertus et al., 2019; Loftus et al., 2018; Plecko & Bareinboim, 2022). This can have severe negative implications: due to unobserved confounding, the notion of causal fairness may be violated, and, as a result, predictions can cause harm as they may be actually *unfair*.

An example is shown in Fig. 1. Here, a prison sentence (=outcome) should be predicted while accounting for the offense and crime history (=observed confounders). Importantly, race (=sensitive attribute) must not affect the sentence. However, many sources of heterogeneity are often missing in the data, such as the housing situation, family background, and birthplace (=unobserved confounders). If not accounted for, the prediction of prison sentences can be *biased* by the confounders and, therefore, *unfair*. Additional examples from credit lending and childcare are in Supplement B.1.

**Our paper:** We analyze the sensitivity of causal fairness to unobserved confounding. Our main contributions are three-fold (see Table 1):[2]

1. We *derive bounds for causal fairness metrics* under different sources of unobserved confounding. Our bounds serve as a refutation strategy to help practitioners examine the sensitivity of machine learning models to unobserved confounding in fairness-critical applications.

2. We *develop a novel neural prediction framework*, which offers worst-case guarantees for the extent to which causal fairness may be violated due to unobserved confounding.

3. We *demonstrate the effectiveness of our framework* in experiments on synthetic and real-world datasets. To the best of our knowledge, we are the first to address causal fairness under unobserved confounding.

## 2 RELATED WORK

**Fairness notions based on causal reasoning:** Fairness notions based on causal reasoning have received increasing attention (e.g., Chiappa, 2019; Chikahara et al., 2021; Huang et al., 2022; Khademi et al., 2019; Kilbertus et al., 2017; 2019; Nabi & Shpitser, 2018; Nabi et al., 2019; Wu et al., 2019a;b; 2022; Yao et al., 2023; Zhang et al., 2017; Zhang & Bareinboim, 2018a;b). For a detailed overview, we refer to (Loftus et al., 2018; Nilforoshan et al., 2022; Plecko & Bareinboim, 2022).

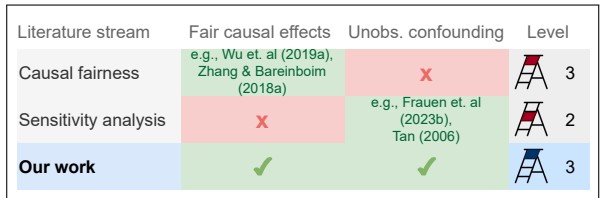

Table 1: We make contributions to multiple literature streams. Level according to Pearl's causality ladder.

Different fairness notions have emerged under the umbrella of causal reasoning (see Supplement C for details). For example, the notion of *counterfactual fairness* (Kusner et al., 2017) makes restrictions on the factual and counterfactual outcomes. A different notion is *causal fairness* (Nabi & Shpitser, 2018; Plecko & Bareinboim, 2022). Causal fairness makes restrictions on the causal graph. It blocks specific causal pathways that influence the outcome variable and are considered unfair. In this paper, we focus on the notion of causal fairness.

**Fairness bounds in non-identifiable settings:** One literature stream (Nabi & Shpitser, 2018; Wu et al., 2019b) establishes fairness bounds for settings that are non-identifiable due to various reasons. While we also derive fairness bounds, both types of bounds serve *different* purposes (i.e., to address general non-identifiability vs. specificity to unobserved confounding).

**Fair predictions that are robust to miss-specification:** The works by Chikahara et al. (2021) and Wu et al. (2019a) aim to learn prediction models that ensure fairness even in the presence of model misspecification. However, none of these works trains prediction models for causal fairness that are robust to unobserved confounding. This is one of our novelties.

**Sensitivity of fairness to unobserved confounding:** To the best of our knowledge, only one work analyzes the sensitivity of fairness to unobserved confounding (Kilbertus et al., 2019), yet it has crucial differences: (1) Their paper focuses on counterfactual fairness, whereas we focus on causal fairness. (2) Their paper allows for unobserved confounding only between covariates, while we follow a fine-grained approach and model different sources of unobserved confounding (e.g., between

---

[2]Both data and code for our framework are available in our GitHub repository.

covariates and sensitive attribute). (3) Their paper builds upon non-linear additive noise models. We do not make such restrictive parametric assumptions.

**Sensitivity models:** Sensitivity models (e.g., Jesson et al., 2022; Rosenbaum, 1987; Tan, 2006) are common tools for partial identification under unobserved confounding. We later adopt sensitivity models to formalize our setting. However, it is not possible to directly apply sensitivity models to our setting (see Table 1). The reason is that sensitivity models perform partial identification on *interventional effects* (=level 2 in Pearl's causality ladder), whereas causal fairness involves (nested) *counterfactual effects* (=level 3). For an in-depth discussion, see Supplement C.3.

**Research gap:** To the best of our knowledge, no work has analyzed the sensitivity of causal fairness to unobserved confounding. We are thus the first to mathematically derive tailored fairness bounds. Using our bounds, we propose a novel framework for learning fair predictions that are robust to the misspecification of the underlying causal graph in terms of unobserved confounding.

## 3 SETUP

**Notation:** We use capital letters to indicate random variables $X$ with realizations $x$. We denote probability distributions over $X$ by $P_X$, i.e., $X \sim P_X$. For ease of notation, we omit the subscript whenever it is clear from the context. If $X$ is discrete, we denote its probability mass function by $P(x) = P(X = x)$ and the conditional probability mass functions by $P(y \mid x) = P(Y = y \mid X = x)$ for a discrete random variable $Y$. If $X$ is continuous, $p(x)$ then denotes the probability density function w.r.t. the Lebesgue measure. A counterfactual outcome to $y$ under intervention $\tilde{x}$ is denoted as $y_{\tilde{x}}$. We denote vectors in bold letters (e.g., $\mathbf{X}$). For a summary of notation, see Supplement A.

We formulate the problem based on the structural causal model (SCM) framework (Pearl, 2014).

**Definition 1** (SCM). *Let* $\mathbf{V} = \{V_1, \dots, V_n\}$ *denote a set of observed endogenous variables, let* $\mathbf{U} \sim P_{\mathbf{U}}$ *denote a set of unobserved exogenous variables, and let* $\mathcal{F} = \{f_{V_1}, \dots, f_{V_n}\}$ *with* $f_{V_i}$ : $Pa(V_i) \subseteq \mathbf{V} \cup \mathbf{U} \to V_i$. *The tuple* $(\mathbf{V}, \mathbf{U}, \mathcal{F}, P_{\mathbf{U}})$ *is called a* structural causal model *(SCM)*.

We assume a directed graph $\mathcal{G}_\mathcal{C}$ induced by SCM $\mathcal{C}$ to be acyclic.

**Setting:** We consider the *Standard Fairness Model* (Plecko & Bareinboim, 2022). The data $\mathbf{V} = \{\mathbf{A}, \mathbf{Z}, \mathbf{M}, Y\}$ consist of a sensitive attribute $\mathbf{A} \in \mathcal{A}$, observed confounders $\mathbf{Z} \in \mathcal{Z}$, mediators $\mathbf{M} \in \mathcal{M}$, and outcome $Y \in \mathbb{R}$.

Unique to our setup is that we allow for three types of unobserved confounding in the model: (1) unobserved confounding between $\mathbf{A}$ and $Y$, which we refer to as *direct unobserved confounding* $U_{\mathrm{DE}}$; (2) between $\mathbf{A}$ and $\mathbf{M}$, called *indirect unobserved confounding* $U_{\mathrm{IE}}$; and (3) between $\mathbf{A}$ and $\mathbf{Z}$, which we refer to as *spurious confounding* $U_{\mathrm{SE}}$. The set of unobserved confounders is then given by $\mathbf{U} = \{U_{\mathrm{DE}}, U_{\mathrm{IE}}, U_{\mathrm{SE}}\}$. Fig. 2 shows the causal graph of the Standard Fairness Model, where we show the different sources of unobserved confounding.

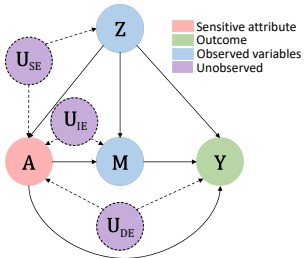

Figure 2: Causal graph of the Standard Fairness Model with different sources of unobserved confounding $U_{\mathrm{DE}}$, $U_{\mathrm{IE}}$, and $U_{\mathrm{SE}}$.

For ease of notation, we focus on discrete variables $\mathbf{Z}, \mathbf{M}$ and binary $\mathbf{A}$, where for simplicity $|\mathbf{A}| = |\mathbf{M}| = |\mathbf{Z}| = 1$. This is common in the literature (e.g., Khademi et al., 2019; Plecko & Bareinboim, 2022; Wu et al., 2019a). Further, it also matches common applications of our framework in law enforcement (Fig. 1), as well as credit lending and childcare (Supplement B.1). In Supplement J, we discuss the extension to continuous features and provide additional experimental results.

**Path-specific causal fairness:** Following Zhang & Bareinboim (2018a), we define the following path-specific causal fairness as: (1) *counterfactual direct effect* (Ctf-DE) for the path $\mathbf{A} \to Y$; (2) *indirect effect* (Ctf-IE) for the path $\mathbf{A} \to \mathbf{M} \to Y$; and (3) *spurious effects* (Ctf-SE) for the path $\mathbf{Z} \to \mathbf{A}$.

Ctf-DE measures direct discrimination based on the sensitive attribute. Ctf-IE and Ctf-SE together measure indirect discrimination through factors related to the sensitive attribute, potentially working as a proxy for the latter. The Ctf-IE corresponds to all indirect causal paths, whereas the Ctf-SE measures all non-causal paths. In the prison sentence example (Fig. 1), the Ctf-DE could measure

unequal prison sentences based on race. The Ctf-IE could be driven by the defendant's court verdict history, which itself might have been illegitimately affected by the offender's race.

**Causal fairness:** Discrimination in terms of causal fairness is formulated through nested counterfactuals [3] for each of the above path-specific causal fairness effects. Formally, one defines the discrimination of realization $A = a_i$ compared $A = a_j$ for each path conditioned on a realization $A = a$, i.e.,

$$\mathrm{DE}_{a_i,a_j}(y \mid a) := P(y_{a_j,m_{a_i}} \mid a) - P(y_{a_i} \mid a), \tag{1}$$

$$\mathrm{IE}_{a_i,a_j}(y \mid a) := P(y_{a_i,m_{a_j}} \mid a) - P(y_{a_i} \mid a), \tag{2}$$

$$\mathrm{SE}_{a_i,a_j}(y) := P(y_{a_i} \mid a_j) - P(y \mid a_i). \tag{3}$$

The total variation, i.e., the overall discrimination of individuals with $A = a_i$ compared to $A = a_j$, can then be explained as $TV_{a_i,a_j} = \mathrm{DE}_{a_i,a_j}(y \mid a_i) - \mathrm{IE}_{a_j,a_i}(y \mid a_i) - \mathrm{SE}_{a_j,a_i}(y)$ (Zhang & Bareinboim, 2018a). Causal fairness is achieved when $\mathrm{DE}_{a_i,a_j}(y \mid a)$, $\mathrm{IE}_{a_i,a_j}(y \mid a)$, $\mathrm{SE}_{a_i,a_j}(y)$ are zero or close to zero. In practice, this is typically operationalized by enforcing that the effects are lower than a user-defined threshold. For ease of notation, we abbreviate the three effects as DE, IE, and SE. We further use CF to refer to any of the above three effects, i.e., $\mathrm{CF} \in \{\mathrm{DE}, \mathrm{IE}, \mathrm{SE}\}$. We note that, although we focus on the three effects specified above, our framework for deriving fairness bounds can also be employed for other fairness notions (see Supplement F).

**Our aim:** We aim to ensure causal fairness of a prediction model in settings where there exists unobserved confounding, i.e., $U_{\mathrm{DE}}, U_{\mathrm{IE}}, U_{\mathrm{SE}} \neq 0$. Existing methods (e.g., Khademi et al., 2019; Zhang & Bareinboim, 2018a) commonly assume that unobserved confounding does *not* exist, i.e., $U_{\mathrm{DE}}, U_{\mathrm{IE}}, U_{\mathrm{SE}} = 0$. Therefore, existing methods are *not* applicable to our setting. Hence, we first derive novel bounds for our setting and then propose a tailored framework.

## 4 BOUNDS FOR CAUSAL FAIRNESS UNDER UNOBSERVED CONFOUNDING

In the presence of unobserved confounding, the different path-specific causal fairness effects – i.e., $\mathrm{CF} \in \{\mathrm{DE}, \mathrm{IE}, \mathrm{SE}\}$ – are *not* identifiable from observational data, and estimates thereof are thus biased (Pearl, 2014). As a remedy, we derive bounds for DE, IE, and SE under unobserved confounding. To this end, we move from point identification to partial identification, where estimate bounds under a sensitivity model $\mathcal{S}$ (elaborated later). Formally, we are interested in an upper bound $\mathrm{CF}_{\mathcal{S}}^+$ and lower bound $\mathrm{CF}_{\mathcal{S}}^-$ corresponding to the path-specific causal fairness effect CF for all $\mathrm{CF} \in \{\mathrm{DE}, \mathrm{IE}, \mathrm{SE}\}$.

To do so, we proceed along the following steps (see Fig. 3): ① We decompose each $\mathrm{CF} \in \{\mathrm{DE}, \mathrm{IE}, \mathrm{SE}\}$ into identifiable parts and unnested interventional effects. ② We then use a suitable sensitivity model $\mathcal{S}$ and the interventional effects to derive the upper $\mathrm{CF}^+$ and lower bound $\mathrm{CF}^-$ for each $\mathrm{CF} \in \{\mathrm{DE}, \mathrm{IE}, \mathrm{SE}\}$. Building upon our bounds, we later develop a neural framework for learning a causally fair prediction model that is robust to unobserved confounding (see Sec. 5).

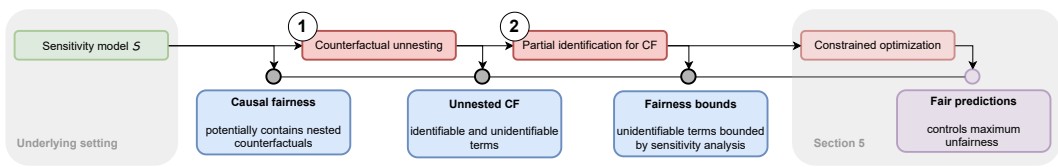

Figure 3: Our workflow for deriving bounds on causal fairness under unobserved confounding.

### 4.1 SETTING: GENERALIZED MARGINAL SENSITIVITY MODEL

We derive fairness bounds in the setting of causal sensitivity analysis (Imbens, 2003; Rosenbaum, 1987). Here, the underlying idea is to relax the assumption of *no* unobserved confounders by allow-

---

[3]Nested counterfactuals express multi-layered hypothetical scenarios by considering counterfactual statements of subsequent events such as $\mathbf{A} \rightarrow \mathbf{M} \rightarrow \mathbf{Y}$. One can view nested counterfactuals as multi-intervention counterfactuals in which one intervention is (partly) intervened on by a second one, e.g., $P(y_{a_i,m_{a_j}})$, which reads as the probability of outcome $Y = y$ under the intervention $A = a_i$ and the intervention $M = m$ where $m$ itself is not affected by intervention $A = a_i$ but $A = a_j$.

ing for a certain strength of unobserved confounding through so-called sensitivity models. In our work, we adopt the generalized marginal sensitivity model (GMSM) (Frauen et al., 2023b). A key benefit of the GMSM is that it can deal with discrete mediators and both discrete and continuous outcomes. Importantly, the GMSM includes many existing sensitivity models and thus allows to derive optimality results for a broad class of models. [4] For an overview of causal sensitivity models and a discussion of their applicability to derive fairness bounds, we refer the reader to Supplement C.2.

**Definition 2** (Generalized marginal sensitivity model (GMSM)). *Let* $\mathbf{V} = \{\mathbf{Z}, \mathbf{A}, \mathbf{M}, Y\}$. *Let* $\mathbf{A}, \mathbf{M}$ *and* $\mathbf{Z}$ *denote a set of observed endogenous variables,* $\mathbf{U}$ *a set of unobserved exogenous variables, and* $\mathcal{G}$ *a causal directed acyclic graph (DAG) on* $\mathbf{V} \cup \mathbf{U}$. *For an observational distribution* $P_{\mathbf{V}}$ *on* $\mathbf{V}$ *and a family* $\mathcal{P}$ *of joint probability distributions on* $\mathbf{V} \cup \mathbf{U}$ *that satisfy*

$$\frac{1}{(1 - \Gamma_W)\, p(a \mid z) + \Gamma_W} \leq \frac{P(U_w = u_W \mid z, a)}{P(U_w = u_W \mid z, \mathrm{do}(a))} \leq \frac{1}{\Gamma_W^{-1}\, p(a \mid z) + \Gamma_W^{-1}} \tag{4}$$

*for* $W \in \{\mathbf{M}, Y\}$, *the tuple* $S = (\mathbf{V}, \mathbf{U}, \mathcal{G}, P_{\mathbf{V}}, \mathcal{P})$ *is called a* weighted generalized marginal sensitivity model *(GMSM) with sensitivity parameter* $\Gamma_W \geq 1$.

The sensitivity parameter $\Gamma_W$ controls the width of the interval defined by the bounds. In practice, it is determined based on domain knowledge of the magnitude of unobserved confounding on the association between $W$ and $\mathbf{A}$ or through data-driven heuristics (e.g., Jin et al., 2023; Kallus et al., 2019). Importantly, our framework applies not only for known $\Gamma_W$ but is also of practical value for unknown $\Gamma_W$. We discuss practical considerations for both cases in Supplement B.2.

We introduce further notation: An acyclic SCM $\mathcal{C} = (\mathbf{V}, \mathbf{U}_\mathcal{C}, \mathcal{F}, P_{\mathbf{U}})$, with $\mathbf{U} \subseteq \mathbf{U}_\mathcal{C}$, $\mathcal{G} \subseteq \mathcal{G}_\mathcal{C}$, is *compatible* with sensitivity model $S$ if $\mathbf{U}_\mathcal{C}$ does not introduce additional unobserved confounding and the probability distribution $P_{\mathbf{V} \cup \mathbf{U}_\mathcal{C}}$ induced by $\mathcal{C}$ belongs to $\mathcal{P}$, i.e., $P_{\mathbf{V} \cup \mathbf{U}_\mathcal{C}} \in \mathcal{P}$.[5] The class of SCMs $\mathcal{C}$ compatible with $S$ is denoted as $\mathcal{K}(S)$.

**Objective for bounding path-specific causal fairness:** We now formalize our objective for identifying $\mathrm{CF}_\mathcal{S}^+$ and $\mathrm{CF}_\mathcal{S}^-$. For $\mathrm{CF}_\mathcal{C} \in \{\mathrm{DE}, \mathrm{IE}, \mathrm{SE}\}$, we aim to find the causal effects that maximize (minimize) $\mathrm{CF}_\mathcal{C}$ over all a possible SCMs compatible with our generalized sensitivity model $\mathcal{S}$, i.e.,

$$\mathrm{CF}_\mathcal{S}^+ = \sup_{\mathcal{C} \in \mathcal{K}(S)} \mathrm{CF}_\mathcal{C} \qquad \text{and} \qquad \mathrm{CF}_\mathcal{S}^- = \inf_{\mathcal{C} \in \mathcal{K}(S)} \mathrm{CF}_\mathcal{C}. \tag{5}$$

As a result, we yield upper $(+)$ and lower $(-)$ bounds $\mathrm{DE}_\mathcal{S}^\pm, \mathrm{IE}_\mathcal{S}^\pm$, and $\mathrm{SE}_\mathcal{S}^\pm$ for the different effects.

### 4.2 Counterfactual unnesting (Step 1)

The path-specific causal fairness effects $\mathrm{CF}_\mathcal{C} \in \{\mathrm{DE}, \mathrm{IE}, \mathrm{SE}\}$ contain nested counterfactuals (=level 3 of Pearl's ladder of causality). Sensitivity analysis, however, is built upon interventional effects (=level 2 of the ladder). Thus, incorporating sensitivity analysis into the fair prediction framework is non-trivial. First, we must decompose the expressions CF into identifiable parts and interventional effects, which solely depend on one intervention at a time. Hence, we now derive an unnested expression of $\mathrm{CF}_\mathcal{C}$ in the following.

**Lemma 1.** $\mathrm{CF}_\mathcal{C} \in \{\mathrm{DE}, \mathrm{IE}, \mathrm{SE}\}$ *can be defined as a monotonically increasing function* $h$ *over a sum of unidentifiable effects* $e \in \mathcal{E}$ *in the SCM* $\mathcal{C}$, *where* $\mathcal{E}$ *denotes the set of all effects in* $\mathcal{C}$, *and an identifiable term* $q$, *i.e.,*

$$\mathrm{CF}_\mathcal{C} = h\Big(\sum_{e \in \mathcal{E}} e(\mathbf{v}, \mathcal{C}) + q\Big) \qquad for \quad \mathrm{CF}_\mathcal{C} \in \{\mathrm{DE}, \mathrm{IE}, \mathrm{SE}\}. \tag{6}$$

*Proof.* The result follows from the ancestral set factorization theorem and the counterfactual factorization theorem in (Correa et al., 2021). A detailed proof is in Supplement D. □

---

[4]Other sensitivity models from the literature such as the marginal sensitivity model (MSM) (Tan, 2006) are special cases of the aforementioned and therefore only of limited use for our general setting (e.g., the MSM can only handle sensitive attributes that are binary but *not* discrete, or they can *not* estimate path-specific causal fairness). In practice, it is up to the user to replace the GMSM with a suitable sensitivity model in our framework.

[5]For a rigorous definition, see Supplement E.

The effects $e$ represent the single-intervention counterfactual effects, which can be bounded through causal sensitivity analysis. The term $q$ can be directly estimated from the data. Now, using Lemma 1, we can rewrite Eq. (5) in an unnested way.

**Remark 1.** *Let $\mathcal{E}^+$ be the set of single-intervention counterfactual effects $e$ with $e(\mathbf{v}, \mathcal{S}) \geq 0$ and $\mathcal{E}^-$ the set of $e$ with $e(\mathbf{v}, \mathcal{S}) < 0$. Then, the bounds for $\mathrm{CF}_{\mathcal{C}} \in \{\mathrm{DE}, \mathrm{IE}, \mathrm{SE}\}$ under GMSM $\mathcal{S}$ in Eq. (5) can be obtained as*

$$\mathrm{CF}_{\mathcal{S}}^+ = h\Big( \sum_{e \in \mathcal{E}^+} \sup_{\mathcal{C} \in \mathcal{K}(S)} e(\mathbf{v}, \mathcal{C}) - \sum_{e \in \mathcal{E}^-} \inf_{\mathcal{C} \in \mathcal{K}(S)} \mid e(\mathbf{v}, \mathcal{C}) \mid +q \Big), \tag{7a}$$

$$\mathrm{CF}_{\mathcal{S}}^- = h\Big( \sum_{e \in \mathcal{E}^+} \inf_{\mathcal{C} \in \mathcal{K}(S)} e(\mathbf{v}, \mathcal{C}) - \sum_{e \in \mathcal{E}^-} \sup_{\mathcal{C} \in \mathcal{K}(S)} \mid e(\mathbf{v}, \mathcal{C}) \mid +q \Big). \tag{7b}$$

## 4.3 SENSITIVITY ANALYSIS TO BOUND PATH-SPECIFIC CAUSAL FAIRNESS (STEP 2)

We now use the expressions from counterfactual unnesting to derive upper $(+)$ and lower $(-)$ bounds for the three path-specific causal effects (i.e., $\mathrm{DE}^\pm$, $\mathrm{IE}^\pm$, and $\mathrm{SE}^\pm$) under unobserved confounding $(\mathbf{U} = \{U_{\mathrm{DE}}, U_{\mathrm{IE}}, U_{\mathrm{SE}}\} \neq 0)$. The following theorem states our main theoretical contribution.

**Theorem 1** (Bounds on path-specific causal fairness). *Let the sensitivity parameters for $M$ and $Y$ in a GMSM be denoted by $\Gamma_M$ and $\Gamma_Y$. For binary sensitive attribute $A \in \{0, 1\}$, where, for simplicity, we focus on discrete $Z$, the individual upper $(+)$ and lower $(-)$ bounds on the path-specific causal fairness effects for specific $y, a_i, a_j$ are given by*

$$
\begin{aligned}
\mathrm{DE}_{a_i, a_j}^\pm(y \mid a_i) = {} & \frac{1}{P(a_i)} \sum_{\substack{z \in \mathbf{Z}, \\ m \in \mathbf{M}}} P^\pm(y \mid m, z, a_j) P^\pm(m \mid z, a_i) P(z) \\
& - \frac{P(a_j)}{P(a_i)} \sum_{\substack{z \in \mathbf{Z}, \\ m \in \mathbf{M}}} P(y \mid m, z, a_j) P(m \mid z, a_i) P(z) - P(y \mid a_i),
\end{aligned}
\tag{8}
$$

$$
\begin{aligned}
\mathrm{IE}_{a_i, a_j}^\pm(y \mid a_j) = {} & \frac{P(a_i)}{P(a_j)} \sum_{\substack{z \in \mathbf{Z}, \\ m \in \mathbf{M}}} P(y \mid m, z, a_i)[P(m \mid z, a_i) - P(m \mid z, a_j)]P(z) \\
& + \sum_{\substack{z \in \mathbf{Z}, \\ m \in \mathbf{M}}} \frac{P(z)}{P(a_j)} \Big( P^\pm(y \mid m, z, a_i) P^\pm(m \mid z, a_j) \\
& \qquad\qquad - P^\mp(y \mid m, z, a_i) P^\mp(m \mid z, a_i) \Big),
\end{aligned}
\tag{9}
$$

$$\mathrm{SE}_{a_i, a_j}^\pm(y) = \frac{1}{P(a_j)} \sum_{\substack{z \in \mathbf{Z}, \\ m \in \mathbf{M}}} P^\pm(y \mid m, z, a_i) P^\pm(m \mid z, a_i) P(z) - \left(1 + \frac{P(a_i)}{P(a_j)}\right) P(y \mid a_i), \tag{10}$$

*for $a_i, a_j \in A$. Note that, in the continuous case, the sum over $Z$ has to be replaced by the integral. For a discrete $M$ with $F(m)$ denoting the CDF of $P(m \mid z, a_i)$, we define*

$$
P^+(m \mid z, a_i) = \begin{cases}
P(m \mid z, a_i)((1 - \Gamma_M^{-1})P(a_i \mid z) + \Gamma_M^{-1}), & \text{if } F(m) < \frac{\Gamma_M}{1 + \Gamma_M} \\
P(m \mid z, a_i)((1 - \Gamma_M)P(a_i \mid z) + \Gamma_M), & \text{if } F(m-1) > \frac{\Gamma_M}{1 + \Gamma_M}, \\
\frac{((1 - P(a_i \mid z))(1 - \Gamma_M)}{1 + \Gamma_M} + F(m)\Gamma_M - F(m-1)\Gamma_M^{-1} \\
\quad + P(a_i \mid z)(F(m)(1 - \Gamma_M) - F(m-1)(1 - \Gamma_M^{-1})), & \text{otherwise,}
\end{cases}
$$

$$
P^-(m \mid z, a_i) = \begin{cases}
P(m \mid z, a_i)((1 - \Gamma_M)P(a_i \mid z) + \Gamma_M), & \text{if } F(m) < \frac{1}{1 + \Gamma_M} \\
P(m \mid z, a_i)((1 - \Gamma_M^{-1})P(a_i \mid z) + \Gamma_M^{-1}), & \text{if } F(m-1) > \frac{1}{1 + \Gamma_M} \\
\frac{(1 - P(a_i \mid z))(1 - \Gamma_M)}{\Gamma_M} + \frac{P(a_i \mid z)}{(1 + \Gamma_M)} + F(m)\Gamma_M^{-1} - F(m-1)\Gamma_M \\
\quad + P(a_i \mid z)(F(m)\Gamma_M^{-1} - F(m-1)(1 - \Gamma_M)) & \text{otherwise.}
\end{cases}
$$

*If $Y$ is discrete, the probability functions $P^+(y \mid m, z, a_i)$ and $P^-(y \mid m, z, a_i)$ are defined analogously. The probability functions for continuous $Y$ are presented in Supplement E.*

*Proof.* We prove the theorem in Supplement D. The proof proceeds as follows: We unnest each counterfactual term in DE, IE, and SE to receive expressions of the form given in Eq. (6). Subsequently, we derive bounds for each unidentifiable part in the unnested counterfactuals through sensitivity analysis. Finally, we combine the bounds to prove the above statement for DE, IE, SE. $\square$

The bounds provide a worst-case estimate for $\text{CF} \in \{\text{DE}, \text{IE}, \text{SE}\}$ for specific $y, a_i, a_j$ under unobserved confounding. Training predictors based on the bounds reduces the risk of harmful and unfair predictions due to violating the no unobserved confounding assumption. The worst-case fairness estimates are guaranteed to contain the true path-specific causal effects for sufficiently large sensitivity parameters. The wider the interval defined through $\text{CF}^+$ and $\text{CF}^-$, the more sensitive the path-specific causal effect to unobserved confounding, i.e., the less confidence we can have in our results if not accounting for unobserved confounding. Thus, one can test the robustness of prediction models as to what extent they are sensitive to unobserved confounding in high-risk applications.

We emphasize that Theorem 1 is independent of the distribution and dimensionality of unobserved confounders and specific SCM formulations. Hence, the bounds are valid for discrete, categorical, and continuous outcome variables, making them widely applicable to real-world problems.

**Remark 2.** *Our bounds are sharp; that is, the bounds from Theorem 1 are optimal, and, therefore, the equality sign in Eq. (5) holds.*

The remark follows from (Frauen et al., 2023b). Our bounds can be interpreted as the supremum or infimum of the path-specific causal effects in any SCM compatible with the sensitivity model.

## 5 BUILDING A FAIR PREDICTION MODEL UNDER UNOBSERVED CONFOUNDING

Our goal is to train a prediction model $f_\theta$ that is fair under unobserved confounding, where fairness is denoted by the path-specific causal fairness effects. Recall that path-specific causal fairness is unidentifiable under unobserved confounding. Therefore, we use our bounds $\text{CF}^+$ and $\text{CF}^-$ for $\text{CF} \in \{\text{DE}, \text{IE}, \text{SE}\}$ from Theorem 1 to limit the worst-case fairness of the prediction, where worst-case refers to the maximum unfairness, i.e., maximum deviation from zero, of each of the path-specific causal effects under a given sensitivity model. Intuitively, we want our prediction model to provide both accurate predictions and fulfill user-defined fairness constraints for each path-specific causal effect, allowing the practitioner to incorporate domain knowledge and business constraints.

**Bound computation:** Theorem 1 requires bounds $P^\pm(y \mid z, m, a)$ and $P^\pm(m \mid z, a)$ to calculate the overall bounds $\text{CF}^+$, $\text{CF}^-$ for $\text{CF} \in \{\text{DE}, \text{IE}, \text{SE}\}$ for specific $y, a_i, a_j$. For training a robustly fair prediction model, we are now interested in deriving effects $\text{CF}_{\mathbb{E}}^+(a_i, a_j)$ of the form $\text{DE}_{a_i, a_j}^\pm(\mathbb{E}[Y] \mid a_i)$ and similarly for IE and SE.[6] As a result, we replace $P^\pm(y \mid z, m, a)$ by the non-random predicted outcome $f_\theta(a, z, m)$. This way, we can estimate the unidentifiable effect $P^\pm(y \mid z, m, a)$ and only need to obtain $P^\pm(m \mid z, a)$. This requires the conditional density estimation for $P(a \mid z)$, $P(m \mid z, a)$ through pre-trained estimators $g_A : \mathbf{Z} \mapsto \mathbf{A}$ and $g_M : \mathbf{A}, \mathbf{Z} \mapsto \mathbf{M}$.

**Objective:** Our objective for training the prediction model $f_\theta$ consists of two aims: (1) a low prediction loss $l(f_\theta)$ and (2) worst-case fairness estimates bounded by a user-defined threshold $\boldsymbol{\gamma} = [\gamma_{\text{DE}}, \gamma_{\text{IE}}, \gamma_{\text{SE}}]^T$. We formulate our objective as a constraint optimization problem:

$$\min_\theta l(f_\theta) \quad \text{s.t.} \max\{|\text{CF}_{\mathbb{E}}^+(a_i, a_j)|, |\text{CF}_{\mathbb{E}}^-(a_i, a_j)|\} \leq \gamma_{\text{CF}}, \quad \forall \, \text{CF} \in \{\text{DE}, \text{IE}, \text{SE}\}, \quad (11)$$

To solve Eq. (11) by gradient-based solvers, we first need to rewrite the constrained optimization problem to a single loss function. For this, we make use of augmented Lagrangian optimization (Nocedal & Wright, 2006), which allows us to add an estimate $\boldsymbol{\lambda}$ of the exact Lagrange multiplier to the objective.[7] We provide pseudocode for our training algorithm in Algorithm 1. In each iteration, we calculate the upper $\text{ub}(\cdot)$ and lower bound $\text{lb}(\cdot)$ on each $\text{CF}_{\mathbb{E}}(a_i, a_j)$, $\text{CF} \in \{\text{DE}, \text{IE}, \text{SE}\}$ according to Theorem 1. In Supplement E, we provide additional explanations and an extended algorithm for multi-class classification.

---

[6]For regression and binary classification, the expectation is a reasonable choice of distribution functional. For multiclass classification, replacing the expectation with other distributional quantiles or constraining the effects for each class separately is also common in practice. More details are in Supplement E.4.

[7]For a detailed description; see Chapter 17 in Nocedal & Wright (2006).

**Algorithm 1:** Training fair prediction models robust to unobserved confounding

**Input:** Data $\{(A_i, Z_i, M_i, Y_i) : i \in \{1, \ldots, n\}\}$, sensitivity parameter $\Gamma_M$, vector of fairness constraints $\boldsymbol{\gamma}$, Lagrangian parameter vectors $\boldsymbol{\lambda}_0$ and $\boldsymbol{\mu}_0$, pre-trained density estimators $g_A, g_M$, initial prediction model $f_{\theta_0}$, convergence criterion $\varepsilon$, Lagrangian update-rate $\alpha$

**Output:** Fair predictions $\{\hat{y}_i : i \in \{1, \ldots, n\}\}$, trained prediction model $f_\theta^*$

1   $f_{\theta_j} \leftarrow f_{\theta_0}$ ; $\boldsymbol{\lambda}_k \leftarrow \boldsymbol{\lambda}_0$ ; $\boldsymbol{\mu}_k \leftarrow \boldsymbol{\mu}_0$ **for** $k \in$ max iterations **do**

2     /* Train prediction model */
    **for** $l \in$ nested epochs **do**

3       $\hat{y} \leftarrow f_{\theta_l}(A, Z, M)$;
      /* Determine $\mathrm{CF}_{\mathbb{E}}^+, \mathrm{CF}_{\mathbb{E}}^-$ for $\mathrm{CF} \in \{\mathrm{DE, IE, SE}\}$ following Theorem 1 */

4       $\mathrm{CF}_{\mathbb{E}}^+ \leftarrow \mathrm{ub}(f_{\theta_l}, (A, Z, M), \Gamma_M, g_A, g_M)$;

5       $\mathrm{CF}_{\mathbb{E}}^- \leftarrow \mathrm{lb}(f_{\theta_l}, (A, Z, M), \Gamma_M, g_A, g_M)$;
      /* Optimization objective following Eq. (11) */

6       **for** $\mathrm{CF} \in \{\mathrm{DE, IE, SE}\}$ **do**

7        $c_{\mathrm{CF}} \leftarrow \max\{|(\mathrm{CF}_{\mathbb{E}}^+)|, |(\mathrm{CF}_{\mathbb{E}}^-)|\}$

8       **end**

9       $\mathbf{c} \leftarrow [c_{\mathrm{DE}}, c_{\mathrm{IE}}, c_{\mathrm{SE}}]^T$;
      lagrangian $\leftarrow \mathrm{loss}(f_{\theta_l}) - \boldsymbol{\lambda}_k(\boldsymbol{\gamma} - \mathbf{c}) - \frac{1}{2\boldsymbol{\mu}_k}(\boldsymbol{\lambda}_k - \boldsymbol{\lambda}_{k-1})^2$;
      /* Update parameters of predictor */

10       $f_{\theta_{l+1}} \leftarrow \mathrm{optimizer}(\mathrm{lagrangian}, f_{\theta_l})$

11     **end**

12     **if** $\mathbf{c} \le \boldsymbol{\gamma}$ **then**
      /* Check for convergence */

13       **if** prediction loss$(f_{\theta_l}) \le \varepsilon$ **then**

14        $f_\theta^* \leftarrow f_{\theta_{l+1}}$;

15        **break**

16       **end**

17     **end**
    /* Update Lagrangian parameters */

18     $\boldsymbol{\lambda}_{k+1} \leftarrow \max\{\boldsymbol{\lambda}_k - \mathbf{c}\boldsymbol{\mu}_k, \mathbf{0}\}$ ; $\boldsymbol{\mu}_{k+1} \leftarrow \alpha\boldsymbol{\mu}_k$

19 **end**

**Implementation details:** We implement the underlying prediction model as a three-layer feed-forward neural network with leaky ReLU activation function and dropout. Our framework additionally requires pre-trained density estimators $g_A, g_M$, for which we also use a feed-forward neural network. Details about the model architectures and hyperparameter tuning are in Supplement G, in which we also report the performance of our prediction models.

## 6 EXPERIMENTS

**Baselines:** We emphasize that baselines for causal fairness under unobserved confounding are absent. We thus compare the following models: (1) **Standard** refers to $f_\theta$ trained only on loss $l$ without causal fairness constraints. Hence, the standard model should lead to unfairness if trained on an unfair dataset, even without unobserved confounding. (2) **Fair naïve** is a prediction model where the loss additionally aims to minimize unfairness in terms of the unconfounded path-specific effects. Hence, causal fairness is considered only under the assumption of no unobserved confounding. (3) **Fair robust** (*ours*) is our framework in Algorithm 1 in which we train the prediction model wrt. causal fairness while additionally accounting for unobserved confounding. Importantly, the architectures of the neural network $f_\theta$ for the different models are identical, so fairness improves only due to our tailored learning objective.

**Performance metrics:** We report path-specific causal fairness by computing bounds on the three effects following Theorem 1. Ideally, the different values remain zero, even in the presence of unobserved confounding. We report the averaged results and the standard deviation over five seeds. We later report results for different sensitivity parameters $\Gamma_M$ in the Supplements.

### 6.1 DATASETS

**Synthetic datasets:** We follow common practice (e.g., Frauen et al., 2023b; Kusner et al., 2017) building upon synthetic datasets for evaluation. This has two key benefits: (1) ground-truth outcomes are available for bench-marking, and (2) we can control the level of unobserved confounding to understand the robustness of our framework.

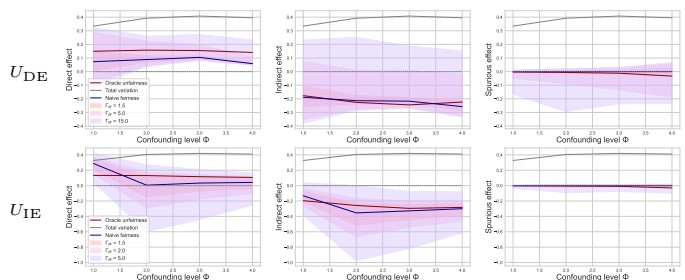

Figure 4: Validity of our bounds. Our bounds successfully contain the oracle effects for different confounding levels $\Phi$. Results for different sensitivity parameters $\Gamma_M$ and $\Gamma_Y = 1$.

We consider two settings with different types of unobserved confounding. Both contain a single binary sensitive attribute, mediator, confounder, and outcome variable. In setting "$U_{\mathrm{DE}}$", we introduce unobserved confounding on the direct effect with confounding level $\Phi$ and in setting "$U_{\mathrm{IE}}$", unobserved confounding on the indirect effect with level $\Phi$. We generate multiple datasets per setting with different confounding levels, which we split into train/val/test sets (60/20/20%). Details are in Supplement H.1.

**Real-world dataset:** Our real-world study is based on the US Survey of Prison Inmates (United States. Bureau of Justice Statistics, 2021). We aim to predict the prison sentence length for drug

offenders. For this, we build upon the causal graph from Fig. 1. We consider race as the sensitive attribute and the prisoner's history as a mediator. Details are in Supplement H.2.

## 6.2 RESULTS

**Results for the synthetic dataset:** We show that our framework generates valid bounds (Fig. 4). Specifically, we demonstrate that our bounds successfully capture the true path-specific causal fairness. We find: (1) Our bounds (red) include the oracle effect as long as the sensitivity parameter is

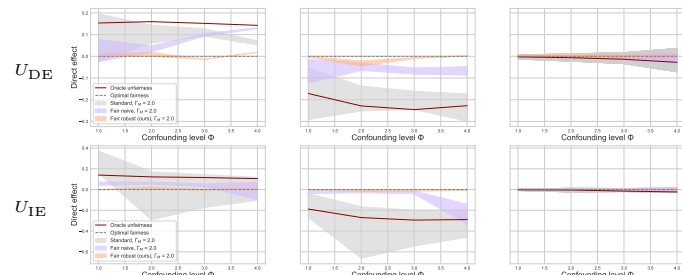

$U_{\mathrm{DE}}$

$U_{\mathrm{IE}}$

Figure 5: Effectiveness of achieving causal fairness as measured by DE, IE, SE (from left to right), which should be close to zero.

sufficiently large. This demonstrates the validity of our bounds. (2) The naïve effects, which assume no unobserved confounding (blue), differ from the oracle effects. This shows that failing to account for unobserved confounding will give wrong estimates of CF $\in \{\mathrm{DE}, \mathrm{IE}, \mathrm{SE}\}$.

We now compare our framework against the baselines (Fig. 5) based on the estimated path-specific causal fairness. We find: (1) The standard model (gray) fails to achieve causal fairness. The path-specific causal fairness effects clearly differ from zero. (2) The fair naïve model (purple) is better than the standard model but also frequently has path-specific causal fairness effects different from zero. (3) Our fair robust model (orange) performs best. The predictions are consistently better in terms of causal fairness. Further, the results have only little variability across different runs, adding to the robustness of our framework (Table 2). Importantly, our framework is also highly effective in dealing with larger confounding levels. See Supplement I for further results.

**Results for the real-world dataset:** We aim to demonstrate the real-world applicability of our framework. Since benchmarking is impossible for real-world data, we seek to provide insights into how our framework operates in practice. In Fig. 6, we compare the predicted prison sentence from (i) the standard model and (ii) our fair robust model. The standard model tends to assign much longer prison sentences to non-white (than to white) offenders, which is due to historical bias and thus deemed unfair. In contrast, our fair robust model assigns prison sentences of similar length.

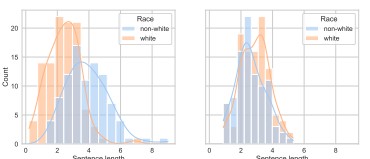

|  |  | DE | IE | SE |
|---|---|---|---|---|
| Standard | upper | $0.14 \pm 0.00$ | $-0.14 \pm 0.00$ | $0.01 \pm 0.00$ |
|  | lower | $0.06 \pm 0.00$ | $-0.25 \pm 0.00$ | $-0.02 \pm 0.00$ |
| Fair naïve | upper | $0.06 \pm 0.05$ | $-0.02 \pm 0.01$ | $0.00 \pm 0.00$ |
|  | lower | $0.05 \pm 0.05$ | $-0.04 \pm 0.02$ | $-0.00 \pm 0.00$ |
| Fair robust (ours) | upper | $0.01 \pm 0.02$ | $-0.01 \pm 0.01$ | $0.00 \pm 0.00$ |
|  | lower | $0.00 \pm 0.02$ | $-0.02 \pm 0.02$ | $0.00 \pm 0.00$ |

Table 2: Estimated bounds on the synthetic dataset "$U_{\mathrm{DE}}$" for confounding level $\Phi = 2$; mean and std. over 5 runs.

Figure 6: Predicted prison sentence length (months) per race group from the standard model (left) and the robustly fair model (right).

## 7 DISCUSSION

**Applicability:** We provide a general framework. First, our framework is directly relevant for many settings in practice (see Supplement B.1). Second, it can be used with both discrete and continuous features (see Supplement J). Third, it is not only applicable to the specific notions above, but it can also be employed for other notions of causal fairness (see Supplement F).

**Conclusion:** Failing to account for unobserved confounding may lead to wrong conclusions about whether a prediction model fulfills causal fairness. First, our framework can be used to perform sensitivity analysis for causal fairness of existing datasets. This is often relevant as information about sensitive attributes (e.g., race) cannot be collected due to privacy laws. Second, our framework can be used to test how robust prediction models under causal fairness constraints are to potential unobserved confounding. To this end, our work is of direct practical value for ensuring the fairness of predictions in high-stakes applications.

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

# A   NOTATION

| | |
|---|---|
| $\mathbf{A}$ | Set of nodes corresponding to sensitive attributes |
| $\mathbf{M}$ | Set of nodes corresponding to mediators |
| $\mathbf{Z}$ | Set of nodes corresponding to observed confounders |
| $\mathbf{U}$ | Set of nodes corresponding to unobserved confounders |
| $U_{\mathrm{DE}}, U_{\mathrm{IE}}, U_{\mathrm{SE}}$ | Unobserved confounders on the direct, indirect and spurious effect, respectively |
| $Y, \hat{Y}$ | Outcome and predicted outcome |
| $\mathcal{X}$ | Set of possible values of random variable $X$ |
| $P(\cdot)$ | Probability distribution over a random variable |
| $p(\cdot)$ | Probability density function of a continuous random variable |
| $\mathcal{P}$ | Family of probability distributions |
| $\mathcal{C}$ | Structural causal model |
| $\mathcal{G}$ | A causal graph |
| $An_{\mathcal{G}}(x_i)$ | Ancestors of $x_i$ in $\mathcal{G}$ |
| $Pa_{\mathcal{G}}(x_i)$ | Parents of $x_i$ in $\mathcal{G}$ |
| $\mathcal{S}$ | Generalized marginal sensitivity model |
| $\Gamma_W$ | Sensitivity parameter |
| $\mathcal{K}(\mathcal{S})$ | Class of SCMs $\mathcal{C}$ compatible with sensitivity model $\mathcal{S}$ |
| $\mathrm{DE}_{a_0,a_i}$ | Counterfactual direct effect of $a_i$ wrt. $a_0$ |
| $\mathrm{DE}_{a_0,a_i}^+, \mathrm{DE}_{a_0,a_i}^-$ | Upper and lower bound on counterfactual direct effect |
| $\mathrm{IE}_{a_0,a_i}$ | Counterfactual indirect effect of $a_i$ wrt. $a_0$ |
| $\mathrm{IE}_{a_0,a_i}^+, \mathrm{IE}_{a_0,a_i}^-$ | Upper and lower bound on counterfactual indirect effect |
| $\mathrm{SE}_{a_0,a_i}$ | Counterfactual spurious effect of $a_i$ wrt. $a_0$ |
| $\mathrm{SE}_{a_0,a_i}^+, \mathrm{SE}_{a_0,a_i}^-$ | Upper and lower bound on counterfactual spurious effect |
| $\mathrm{CF}, \mathrm{CF}^+, \mathrm{CF}^-$ | Causal fairness notion with upper and lower bound |
| $\mathcal{D}$ | Functional mapping a density to a scalar value |
| $f_\theta$ | Prediction model with parameters $\theta$ |
| $g$ | Density estimator |
| $\gamma$ | Fairness constraint |
| $\Phi$ | Confounding level |
| $\mathcal{N}(\boldsymbol{\mu}, \boldsymbol{\Sigma})$ | Gaussian distribution with mean $\boldsymbol{\mu}$ and covariance $\boldsymbol{\Sigma}$ |

## B    DISCUSSION OF APPLICATIONS

### B.1    EXAMPLES: CAUSAL FAIRNESS WITH UNOBSERVED CONFOUNDING

**Prediction of credit default risk:** When applying for a loan, banks commonly evaluate the applicant's credit default risk based on various factors, such as the applicant's credit history, occupation, and the presence of a guarantor. Following the goal of profit maximization and risk minimization, the bank will only grant the loan to solvent applicants, i.e., applicants with a low default risk.

Various fair lending laws across the globe require that the risk assessment must be fair and, to this end, should not be affected by sensitive information. An example could be whether or not the applicant is an immigrant worker. The example is shown in Figure 7. Therein, the default risk is estimated based on information about the occupation and wage of the applicant, the size of the loan, the presence of a guarantor,

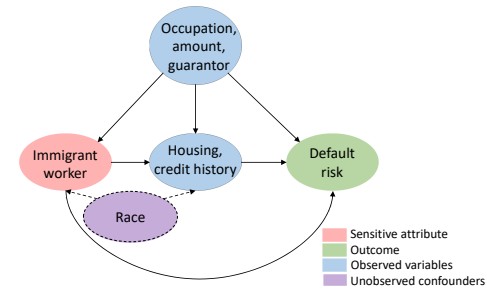

Figure 7: Example: causal graph for predicting credit default risk.

the housing situation of the applicant (since hypothecary credits can be based on the property), and the credit history. A fair and strategy-optimizing prediction model should (i) control for any direct, indirect, and spurious effect stemming from the fact of an applicant being an immigrant worker and (ii) incorporate the potential effect of the unobserved variable race on the immigration as well as credit history. Of note, laws in various countries forbid the collection and storage of information about race so that such information is often missing, and, therefore, it presents an unobserved confounder.

Training a machine learning method for predicting a new applicant's default risk without accounting for fairness constraints and potential unobserved confounding will likely result in biased and unfair predictions and, thus, biased loan allocation. In the stated example, immigrant workers might not receive a loan, although they would constitute a good risk in an unbiased assessment, whereas nationals with a high default risk might falsely be granted a loan. This not only reinforces the societal bias present in the data but also fails to optimize the bank's strategy in terms of profit maximization and risk minimization.

**Prediction of acceptance of child to nursery:** Several works aim to leverage machine learning models in public organizations. One example is the following. Due to a high birth rate and a shortage of available places at childcare facilities, not all children can often be admitted to a nursery. The decision of which child to admit might depend on the parents occupation, the number of children in the family, and the housing situation. The causal diagram is shown in Figure 8, where we detail the mechanisms behind a recommendation for admission to the nursery. In the present example, we argue that the financial status of the family should only have a limited effect on the recommendation of admission, i.e., the prediction unfairness introduced by the financial status should be lower than a specific threshold. Furthermore, multiple unobserved factors typically influence the

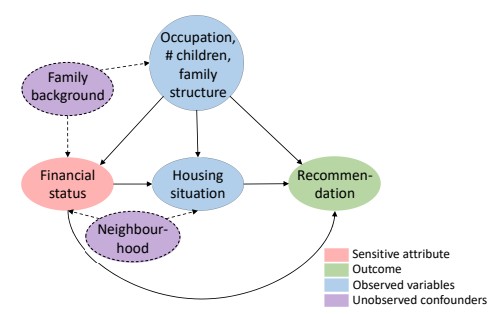

Figure 8: Example: causal graph for predicting acceptance to the nursery.

family's financial status and the observed variables. For example, the financial status and occupation could both be influenced by the family background, potentially mediated through the parent's

education. The residential neighborhood is related to the family's financial status and the housing situation. Then, a good financial standing might indicate the family's ability to pay for personalized private childcare (e.g., a nanny) and, therefore, could have a certain effect on the recommendation for admission to a nursery. However, a machine learning model trained solely on the observed data will inevitably provide biased and unfair recommendations.

The above examples show two important applications of causal fairness in practice where unobserved confounding is prevalent and could be addressed by our framework.

### B.2 DISCUSSION ABOUT KNOWLEDGE OF THE STRENGTH OF UNOBSERVED CONFOUNDING

In practice, knowledge of the unobserved confounding (UC) strength is beneficial but not fully necessary. Importantly, our framework is of direct help in practice when (i) the maximum UC strength is known or even when (ii) unknown. We discuss both use cases below:

1. There are often good reasons why the UC strength is known or can be upper-bounded. Many fairness-relevant applications are rooted in social science, where there is a good understanding of potential causes for biases (e.g., around gender, age, etc.). Hence, it is often reasonable to assume that the UC strength is in a specific relationship with some other observed variable. For example, based on domain knowledge, one can often say that the UC strength is not as strong as (a multiple of) the observed variable (e.g., Cinelli & Hazlett, 2020; Franks et al., 2019; Masten & Poirier, 2020; Zhang et al., 2016). In such cases, the UC strength $\Gamma_W$ can be set by practitioners accordingly, and our framework can be applied as discussed in the main paper

2. Even when the UC strength is unknown, our framework is of significant value in practice. In this case, practitioners can employ our bounds to test the sensitivity of causal fairness notions with respect to unobserved confounding in the data. One way to do so is to use our framework to calculate bounds for increasing sensitivity parameters until the interval defined by the bounds contains a certain value of interest, e.g., zero (indicating complete fairness). The minimal sensitivity parameter to achieve this goal corresponds to the level of unobserved confounding, which would be necessary to consider the data fair. Hence, one can also view the sensitivity parameter as our uncertainty about the fairness of our prediction model. Smaller sensitivity parameters that do not achieve the goal still provide information about the sign, i.e., direction, of the unfair effect Hsu & Small (2013). Generally, testing various sensitivity parameters is a common technique in real-world applications of sensitivity analysis (e.g., Hsu & Small, 2013; Jin et al., 2023; VanderWeele & Ding, 2017). In sum, even when the UC strength is unknown, our framework can thus be of large practical value.

Furthermore, several approaches have been proposed to choose sensitivity parameters in practice. We refer to, e.g., Jin et al. (2023); Kallus et al. (2019); Díaz & van der Laan (2013); Imai & Yamamoto (2013) for further discussions.

## C  EXTENDED RELATED WORK

### C.1  FAIRNESS IN MACHINE LEARNING

In the following, we present an extended discussion of related works. First, we give a brief taxonomy of algorithmic fairness. Then, we discuss related causal fairness notions and explain the difference between counterfactual and causal fairness. For the latter, we heavily rely on Plecko & Bareinboim (2022) and refer to their work for an in-depth discussion of causal fairness analysis.

**Algorithmic fairness:** There are mainly two classes of fairness notions, namely, statistical notions and notions based on causal reasoning (Fawkes et al., 2022). Often, statistical fairness notions are incompatible with one another (Fawkes et al., 2022; Kilbertus et al., 2019; Kleinberg et al., 2017) and cannot provide intuitive results (Nabi & Shpitser, 2018). Causality-based notions can help in overcoming such problems. Relatedly, fairness notions can be defined on a group or individual level, where the specific choice depends on practical considerations (Loftus et al., 2018). Our framework focuses on group-level fairness. Although developed for training fair prediction models, the results can also be used to explain individual discrimination in a prediction model.

At a technical level, fairness in predictions can be achieved at different steps: during pre-processing, in-processing, or post-processing. Since a prediction model trained on data fair labels is not necessarily fair itself (Ashurst et al., 2022), pre-processing fairness notions do not always mitigate the unfairness present in the resulting prediction. Therefore, in our work, we seek to ensure fairness in the presence of unobserved confounding through an in-processing approach.

**Causal vs. counterfactual fairness:** Causal fairness measures quantify the association of a sensitive attribute and the target variable through causal mechanisms in a structural causal model (SCM) (Plecko & Bareinboim, 2022). Specifically, a causal fairness notion must fulfill the three elementary structural fairness criteria: (1) the structural direct criterion, assessing if the target is a function of the sensitive attribute; (2) the structural indirect criterion, determining the presence of an effect of a mediator variable on the target which in turn is influenced by the sensitive attribute; and (3) the structural spurious criterion, evaluating the presence of a confounder between the sensitive attribute and the target. Causal fairness notions can, therefore, differentiate between direct and indirect discrimination to mathematically account for legal definitions of fairness, such as *disparate impact* and *disparate treatment*, or incorporate business necessities.

As a specific notion, counterfactual fairness (Kusner et al., 2017) enforces the outcome variable to be identical in both the current observation and a hypothetical counterfactual world in which the protected attribute has a different realization. It specifically focuses on the total variation in the outcome caused by a change in the sensitive attribute and does not differentiate between multiple pathways of discrimination. Many extensions and methods for ensuring counterfactual fairness have been introduced in the literature (e.g., Chiappa, 2019; Kilbertus et al., 2017; Ma et al., 2023).

Several causal fairness measures are also based on counterfactuals (see Plecko & Bareinboim (2022) for a taxonomy overview), so we warn that the naming in the literature may be misleading. Therefore, we highlight the differences in the following. Importantly, counterfactual fairness and any fairness notion built upon it differs from counterfactual-based causal fairness notions in three fundamental aspects: (1) *admissibility*, (2) *ancestral closure*, and (3) *identifiability*. We discuss the aspects in the following:

1. Admissibility: Even if the counterfactual fairness metric evaluates to zero and thus implies that a model is fair, there is still no guarantee for both the direct and indirect association of the sensitive attribute and the target being zero, as the effects might cancel out. To this end, counterfactual fairness is inadmissible w.r.t. the structural criteria.

2. Ancestral closure: Ancestral closure requires all ancestors of the sensitive attribute to be observed. This implies that there cannot be any unobserved confounders on an association containing the sensitive attribute, which is unlikely to hold in practice. Causal fairness notions, on the other hand, allow for endogenous ancestors.

3. Identifiability: Throughout our paper, we have dealt with unidentifiability of causal fairness due to unobserved confounding, arguing that the assumption of no unobserved confounding does not hold in practice. If the full SCM was known, the fairness notion would be identifiable. Counterfactual fairness, however, is never identifiable from observational data

> if mediators between the sensitive attribute and the target exist, even if the full SCM is specified.

In sum, both causal vs. counterfactual fairness are entirely different notions. In our work, we focus on causal fairness.

## C.2 CAUSAL SENSITIVITY ANALYSIS

Causal sensitivity analysis has widely been employed for partial identification problems. By imposing assumptions on the strength of unobserved confounding and, therefore, relaxing the assumption of no unobserved confounding, various causal effects can be bounded in the presence of unobserved confounding. The resulting bounds can often prove that the causal quantities of interest cannot be explained away by unobserved confounding. Especially for real-world studies and policy learning, sensitivity models thus provide important tools for assessing the sensitivity of a causal estimate to unobserved confounding through deriving informative regions for the causal effect (Kallus et al., 2019).

The main sensitivity models introduced in the literature are: the sensitivity model of Rosenbaum (Rosenbaum, 1987) employing randomization tests; the non-parametric marginal sensitivity model (Tan, 2006); and the recent $f$-sensitivity models (Jin et al., 2023). For binary treatments $A \in \{0, 1\}$, the marginal sensitivity model (MSM) Tan (2006) is defined as

$$\frac{1}{\Gamma} \leq \frac{\pi(z)}{1 - \pi(z)} \frac{1 - \pi(z, u)}{\pi(z, u)} \leq \Gamma, \tag{12}$$

where $\pi(z) = P(A = 1 \mid z)$ denotes the observed propensity score and $\pi(z, u) = P(A = 1 \mid z, u)$ denotes the full propensity score.

Multiple extensions have been developed recently, especially for the MSM, which has received much attention in the literature. Most work has focused on the average treatment effect (ATE) and conditional average treatment effect (CATE) for binary treatment settings (e.g., Dorn & Guo, 2022; Dorn et al., 2022; Jesson et al., 2021; Kallus et al., 2019; Oprescu et al., 2023; Zhao et al., 2019). Popular extensions for the MSM include sensitivity models for continuous treatments (Jesson et al., 2022; Marmarelis et al., 2023a). For continuous treatments $A$, the continuous marginal sensitivity model (CMSM) Jesson et al. (2022) is defined through

$$\frac{1}{\Gamma} \leq \frac{P(a \mid z, u)}{P(a \mid z)} \leq \Gamma. \tag{13}$$

Other extensions include augmentations to individual treatment effects (Yin et al., 2022; Jin et al., 2023; Marmarelis et al., 2023b).

The work by Frauen et al. (2023b) introduces the Generalized Marginal Sensitivity Model (GMSM): For an observational distribution $P_{\mathbf{V}}$ on $\mathbf{V}$ and a family $\mathcal{P}$ of joint probability distributions on $\mathbf{V} \cup \mathbf{U}$ that satisfy

$$\frac{1}{(1 - \Gamma_W) \, q_W(a, z) + \Gamma_W} \leq \frac{P(U_w = u_W \mid z, a)}{P(U_w = u_W \mid z, \mathrm{do}(a))} \leq \frac{1}{\Gamma_W^{-1} \, q_W(a, z) + \Gamma_W^{-1}} \tag{14}$$

for $W \in \{M, Y\}$ and weight function $q_W(a, z) \in [0, 1]$, the tuple $S = (\mathbf{V}, \mathbf{U}, \mathcal{G}, P_{\mathbf{V}}, \mathcal{P})$ is called a *weighted generalized marginal sensitivity model* (GMSM) with sensitivity parameter $\Gamma_W \geq 1$. It can be shown that, for weight functions $q(a, z) = P(a \mid z)$ (MSM) and $q(a, z) = 0$, the MSM and the CMSM are special cases of the weighted GMSM (Frauen et al., 2023b). Therefore, the GMSM generalizes many of the aforementioned approaches as it allows to derive bounds for binary and continuous treatments as well as different causal queries (e.g., CATE, distributional effects).

Causal sensitivity analysis has also been applied to other settings, such as off-policy learning (e.g., Hatt et al., 2022; Kallus & Zhou, 2018) or partial identification of counterfactual queries (Melnychuk et al., 2023). Recently, neural frameworks for automated generalized sensitivity analysis have been proposed (Frauen et al., 2024). Only one other work besides us has employed sensitivity analysis to study fairness notions under unobserved confounding (Kilbertus et al., 2019). Nevertheless, this work focuses on the notion of counterfactual fairness (and *not* causal fairness) and is limited to non-linear additive noise models.

**Rationale behind our choice of sensitivity models:**

In the following, we compare multiple extensions of the MSM with respect to applicability and discuss the strengths and weaknesses. Thereby, we provide a justification of why we adopt the GMSM in our paper.

A general benefit of the MSM is that it does not impose parametric assumptions on the data-generating process, thus enabling wide applicability to many domains. Nevertheless, the sensitivity model is only defined for a single binary treatment variable. Mutliple extensions derive bounds based on the MSM (e.g., Dorn & Guo, 2022; Dorn et al., 2022; Jesson et al., 2021; Kallus et al., 2019; Oprescu et al., 2023; Zhao et al., 2019). However, the bounds provided by the extensions above are unnecessarily conservative. Therefore, Dorn & Guo (2022) and Jin et al. (2023) derived closed-form solutions for *sharp* bounds under the MSM.

The main weaknesses of the above extensions are (i) the limitation to binary treatment and (ii) the restricted focus on the (conditional) average treatment effect. To overcome weakness (i), Bonvini et al. (2022); Jesson et al. (2022) and Marmarelis et al. (2023a) proposed extensions for continuous treatments. Additionally, Bonvini et al. (2022) extended the setting to time-varying treatment and confounding variables. To overcome weakness (ii), Jin et al. (2023); Marmarelis et al. (2023b) and Yin et al. (2022) introduced sensitivity models to cover the individual treatment effect (ITE).

In our work, we adopt the GMSM. The generalized marginal sensitivity model (GMSM) (Frauen et al., 2023b) provides a general causal sensitivity framework that can incorporate continuous, discrete, and time-varying treatments. The resulting bounds are *sharp* and applicable to multiple causal effects (e.g., CATE, ATE) as well as to mediation analysis, path analysis, and for distributional effects. Overall, the GMSM is thus highly suitable for deriving bounds on causal fairness metrics and thus makes our framework widely applicable to various settings (e.g., discrete and continuous variables, etc.). Of note, we use the GMSM only to formalize our setting, while the actual derivation of bounds is *non-trivial* (and is *not* a direction application of the GMSM). The reason is that the GMSM makes interventional queries (=level 2 in Pearl's causality ladder), while our task involves counterfactual queries (=level 3). Hence, existing bounds are *not* applicable; instead, a new and careful derivation of bounds that are tailored to our setting is needed.

We also summarize the above discussion in Table 3, which provides a systematic overview of the applicability of existing MSM extensions.

Table 3: Overview of key extensions of the MSM for causal sensitivity analyses. Applicability/non-applicability is indicated by a green tick (✓) and a red cross (✗), respectively.

| MSM extensions | Treatment type | | Causal query | | |
|---|---|---|---|---|---|
| | Binary | Cont. | (C)ATE | Distributional effects | Individual treatment effect (ITE) |
| Tan (2006) | ✓ | ✗ | ✓ | ✗ | ✗ |
| Kallus et al. (2019) | ✓ | ✗ | ✓ | ✗ | ✗ |
| Zhao et al. (2019) | ✓ | ✗ | ✓ | ✗ | ✗ |
| Jesson et al. (2021) | ✓ | ✗ | ✓ | ✗ | ✗ |
| Dorn & Guo (2022) | ✓ | ✗ | ✓ | ✗ | ✗ |
| Dorn et al. (2022) | ✓ | ✗ | ✓ | ✗ | ✗ |
| Oprescu et al. (2023) | ✓ | ✗ | ✓ | ✗ | ✗ |
| Soriano et al. (2023) | ✓ | ✗ | ✓ | ✗ | ✗ |
| Bonvini et al. (2022) | ✗ | ✓ | ✓ | ✗ | ✗ |
| Jesson et al. (2022) | ✗ | ✓ | ✓ | ✗ | ✗ |
| Marmarelis et al. (2023a) | ✗ | ✓ | ✓ | ✗ | ✗ |
| Jin et al. (2023) | ✓ | ✗ | ✗ | ✗ | ✓ |
| Yin et al. (2022) | ✓ | ✗ | ✗ | ✗ | ✓ |
| Marmarelis et al. (2023b) | ✓ | ✗ | ✗ | ✗ | ✓ |
| Frauen et al. (2023b) | ✓ | ✓ | ✓ | ✓ | ✓ |

## C.3 CHALLENGES IN SENSITIVITY ANALYSIS FOR CAUSAL FAIRNESS

We transfer concepts from sensitivity analysis to the causal fairness literature by (i) showing the applicability of sensitivity analysis outside the field of standard causal effects, and (ii) providing a solution tool for assessing causal fairness under unobserved confounding as well as a direction for future research. Nevertheless, sensitivity models are *not* directly compatible with causal fairness notions. We elaborate on the difficulty and our proposed solution below.

In the following, we rephrase our workflow in terms of Pearl's ladder of causality (Fig. 9): We aim to provide bounds for causal fairness notions, which can be estimated from the data. *Causal fairness* notions are located on *level three* of Pearl's ladder, i.e., they contain counterfactual expressions. However, *sensitivity models* are interventional queries and thus located on *level two*. Hence, existing bounds from sensitivity models are *not* applicable. To remedy the above, we need to develop a framework to bridge the gap between levels three and two.

Therefore, we propose the following approach to deriving bounds: (i) we propose to decompose the nested counterfactuals into interventional (non-identifiable due to unobserved confounding) and identifiable effects. This step is non-trivial and requires customization for every causal fairness notion. Then, (ii) we employ sensitivity analysis to derive bounds on the interventional terms. The resulting fairness bounds consist of a concatenation of sensitivity bounds and the decomposed identifiable effects. We present a schematic of our workflow in terms of the ladder of causality in Fig. 9.

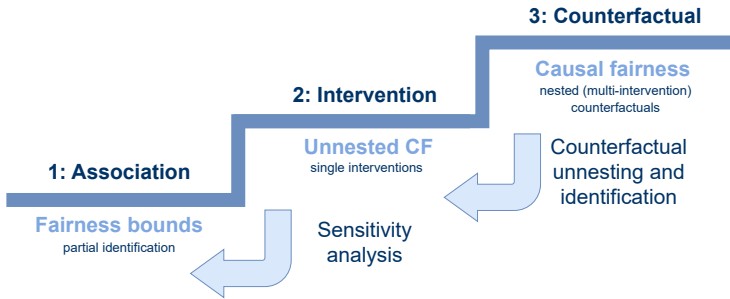

Figure 9: Our workflow in terms of Pearl's ladder of causality. Note: sensitivity models like the GMSM are interventional queries (=level 2 in Pearl's causality ladder), while our task involves counterfactual queries (=level 3), because of which a tailored framework for deriving bounds is needed.

# D PROOFS

In the following, we derive proofs for our main theorem from Section 4. According to the three-step approach presented in Section 4, we first provide a theoretical background on the sensitivity model $\mathcal{S}$ (the GMSM). Then, we perform counterfactual unnesting of the path-specific causal effects. Finally, we prove the overall bounds per counterfactual effect.

## D.1 THEORETICAL BACKGROUND ON GMSMs

We can derive bounds on single-counterfactual causal effects, i.e. unnested counterfactuals, in a GMSM based on the following theorem.

**Theorem 2.** *(Frauen et al., 2023b) Let S be a GMSM with sensitivity parameters $\Gamma_W$, $W \in \mathbf{M} \cup \{Y\}$. Further, we restrict each unobserved confounder $U \in \mathbf{U}$ to be the parent of only one element in $\mathbf{M} \cup \{Y\}$, i.e., there does not exist unobserved confounding between mediators and outcome as well as between mediators themselves. Let $\mathcal{K}(S)$ denote the class of SCMs C compatible with S. Let $F(\cdot)$ as denote the CDF corresponding to $P(\cdot \mid Z, M_W, A)$. For a continuous W, we define*

$$P^+(w \mid z, m_W, a) = \begin{cases} \frac{1}{s_W^+} P(w \mid z, m_W, a), & \text{if } F(w) \leq \frac{\Gamma_W}{1+\Gamma_W} \\ \frac{1}{s_W^-} P(w \mid z, m_W, a), & \text{if } F(w) > \frac{\Gamma_W}{1+\Gamma_W} \end{cases} \tag{15}$$

*and*

$$P^-(w \mid z, m_W, a) = \begin{cases} \frac{1}{s_W^-} P(w \mid z, m_W, a), & \text{if } F(w) \leq \frac{1}{1+\Gamma_W} \\ \frac{1}{s_W^+} P(w \mid z, m_W, a), & \text{if } F(w) > \frac{1}{1+\Gamma_W}. \end{cases} \tag{16}$$

*For a discrete $W \in \mathbb{N}$, we define*

$$P^+(w \mid z, m_W, a) = \begin{cases} \frac{1}{s_W^+} P(w \mid z, m_W, a), & \text{if } F(w) < \frac{\Gamma_W}{1+\Gamma_W} \\ \frac{1}{s_W^-} P(w \mid z, m_W, a), & \text{if } F(w-1) > \frac{\Gamma_W}{1+\Gamma_W} \\ \frac{1}{s_W^+}(\frac{\Gamma_W}{1+\Gamma_W} - F(w-1)) + \frac{1}{s_W^-}(F(w) - \frac{\Gamma_W}{1+\Gamma_W}), & \text{otherwise.} \end{cases} \tag{17}$$

*and*

$$P^-(w \mid z, m_W, a) = \begin{cases} \frac{1}{s_W^-} P(w \mid z, m_W, a), & \text{if } F(w) < \frac{1}{1+\Gamma_W} \\ \frac{1}{s_W^+} P(w \mid z, m_W, a), & \text{if } F(w-1) > \frac{1}{1+\Gamma_W} \\ \frac{1}{s_W^-}(\frac{1}{1+\Gamma_W} - F(w-1)) + \frac{1}{s_W^+}(F(w) - \frac{1}{1+\Gamma_W}), & \text{otherwise.} \end{cases} \tag{18}$$

*With $F^+(\cdot)$, $F^-(\cdot)$ and $F_{\mathcal{C}}(\cdot)$ denoting the conditional CDFs corresponding to $P^+(w \mid z, m_W, a)$, $P^-(w \mid z, m_W, a)$ and $P_{\mathcal{C}}(\cdot \mid z, m_W, \mathrm{do}(A = a))$, respectively, we yield*

$$F^+(\cdot) = \inf_{\mathcal{C} \in \mathcal{K}(S)} F_{\mathcal{C}}(w) \qquad \text{and} \qquad F^-(\cdot) = \sup_{\mathcal{C} \in \mathcal{K}(S)} F_{\mathcal{C}}(w) \tag{19}$$

*for all w.*

## D.2 COUNTERFACTUAL UNNESTING

We perform counterfactual unnesting based on the counterfactual identifiability theory presented in (Correa et al., 2021). We adopt the notation therein. Let $V_{\mathbf{X}}$, $V \in \mathbf{V}$, $\mathbf{X} \subseteq \mathbf{V}$ denote a counterfactual expression for $V$. The ancestral set of $V_{\mathbf{X}}$ is given by $An_{\mathcal{G}}(V_{\mathbf{X}}) = \{W_{\mathbf{z}} \mid W \in An_{\mathcal{G}_{\underline{\mathbf{X}}}}(V), \mathbf{z} = \mathbf{X} \cap An_{\mathcal{G}_{\underline{\mathbf{X}}}}(V)\}$, where $\mathcal{G}_{\underline{\mathbf{X}}}$ denotes the graph deduced from $\mathcal{G}$ when removing all edges going out of variables $\mathbf{X}$.

For the general case, let $\mathbf{W}_*$ denote a set of arbitrary counterfactual variables. The corresponding ancestral set is defined as $An_{\mathcal{G}}(\mathbf{W}_*) = \bigcup_{W_t \in \mathbf{W}_*} An_{\mathcal{G}}(W_t)$, where $t$ denotes the respective intervention.

**Theorem 3** (Ancestral set factorization from (Correa et al., 2021)). *Let $\boldsymbol{W}_*$ be an ancestral set, i.e., $An_{\mathcal{G}}(\boldsymbol{W}_*) = \boldsymbol{W}_*$, and let $\boldsymbol{w}_*$ be a realization of $\boldsymbol{W}_*$. Then,*

$$P(\boldsymbol{W}_* = \boldsymbol{w}_*) = P\left(\bigwedge_{W_t \in \boldsymbol{W}_*} W_{\boldsymbol{pa}(W)} = w_t\right), \tag{20}$$

*where the subscript $\boldsymbol{t}$ denotes the set of interventions on $W$ present in $\boldsymbol{W}_*$. Note that $\boldsymbol{t}$ might be the empty set, i.e., the variable $W$ is not intervened on. The set of interventions $\boldsymbol{pa}(W)$ for each $W_{\boldsymbol{t}} \in \boldsymbol{W}_*$ corresponds to the interventions $\boldsymbol{t}$ for non-empty $\boldsymbol{t}$. Otherwise, $\boldsymbol{pa}(W)$ is given by the parents $Pa_{\mathcal{G}}(W)$ of $W$.*

Analyzing counterfactuals such as in Eq. (20) based on their counterfactual factors ('ctf-factors') can aid in deciding its identifiability in a given graph. A counterfactual factor $P(W_{1_{[\boldsymbol{pa}(W_1)]}} = w_1, W_{2_{[\boldsymbol{pa}(W_2)]}} = w_2, \ldots, W_{l_{[\boldsymbol{pa}(W_l)]}} = w_l)$ for $W_i \in \boldsymbol{V}$, where potentially $W_i = W_j$ for some $i, j \in \{1, \ldots, l\}$ generalizes the parent-child relationship encoded in c-factors to the counterfactual domain. The brackets around the subscript denote the interventions on the enumerated variables $W_i$ for $i \in \{1, \ldots, l\}$.

**Theorem 4** (Counterfactual factorization from (Correa et al., 2021)). *Let $P(\boldsymbol{W}_* = \boldsymbol{w}_*)$ be a counterfactual factor with a topological order over the variables in $\mathcal{G}[\boldsymbol{V}(\boldsymbol{W}_*)]$. Further, let $\boldsymbol{C}_1, \ldots, \boldsymbol{C}_k$ be the c-components of the stated graph, $\boldsymbol{C}_{j*} := \{W_{\boldsymbol{pa}(W)} \in \boldsymbol{W}_* \mid W \in \boldsymbol{C}_j\}$ with $\boldsymbol{c}_j$ the values in $\boldsymbol{w}_*$. Then, one yields*

$$P(\boldsymbol{W}_* = \boldsymbol{w}_*) = \prod_j P\left(\boldsymbol{C}_{j*} = \boldsymbol{c}_{j*}\right). \tag{21}$$

In the following, we will use the stated counterfactual unnesting theory to split the expressions $P(y_{a_i} \mid a)$ and $P(y_{a_i, m_{a_j}} \mid a)$ into the respective counterfactual factors. We thus obtain

$$P(y_{a_i} \mid a) \stackrel{(*)}{=} \frac{1}{P(a)} \sum_{\substack{z \in \mathbf{Z}, \\ m \in \mathbf{M}}} P(y_{a_i}, a, z, m_{a_i}) \stackrel{(**)}{=} \frac{1}{P(a)} \sum_{\substack{z \in \mathbf{Z}, \\ m \in \mathbf{M}}} P(y_{a_i, z, m}, a_z, z, m_{a_i, z}), \tag{22}$$

where we leverage the ancestral set expansion $(*)$ and the counterfactual factorization $(**)$ from above, respectively. If we assume there exists unobserved confounding between sensitive attribute and mediator ($U_{\text{IE}}$), sensitive attribute and covariates ($U_{\text{SE}}$), and sensitive attribute and outcome ($U_{\text{DE}}$), this expression cannot be reduced further.

The expression for the second term of interest follows similarly via

$$P(y_{a_i, m_{a_j}} \mid a) = \frac{1}{P(a)} \sum_{m \in \mathbf{M}} P(y_{a_i, m}, a, m_{a_j}) = \frac{1}{P(a)} \sum_{\substack{z \in \mathbf{Z}, \\ m \in \mathbf{M}}} P(y_{a_i, z, m}, a_z, z, m_{a_j, z}). \tag{23}$$

As a result, we can write path-specific causal fairness as a combination of identifiable effects and single-counterfactual effects, which subsequently can be bounded following Theorem 2. Hence, we yield

$$\text{DE}_{a_i, a_j}(y \mid a_i) = P(y_{a_j, m_{a_i}} \mid a_i) - P(y_{a_i} \mid a_i) \tag{24}$$

$$= \frac{1}{P(a_i)} \sum_{\substack{z \in \mathbf{Z}, \\ m \in \mathbf{M}}} P(y_{a_j, z, m}, a_z, z, m_{a_i, z}) - P(y \mid a_i) \tag{25}$$

$$= \frac{1}{P(a_i)} \sum_{\substack{z \in \mathbf{Z}, \\ m \in \mathbf{M}}} P(y|z, m, do(a_j)) P(m \mid do(a_i), z) P(z) \tag{26}$$

$$- \frac{P(a_j)}{P(a_i)} \sum_{\substack{z \in \mathbf{Z}, \\ m \in \mathbf{M}}} P(y \mid m, z, a_j) P(m, z \mid a_i) - P(y \mid a_i),$$

$$\text{IE}_{a_i,a_j}(y \mid a_j) = P(y_{a_i,m_{a_j}} \mid a_j) - P(y_{a_i} \mid a_j) \tag{27}$$

$$= \frac{1}{P(a_j)} \Big( \sum_{\substack{z \in \mathbf{Z}, \\ m \in \mathbf{M}}} P(y_{a_i,z,m}, a_z, z, m_{a_j,z}) - \sum_{\substack{z \in \mathbf{Z}, \\ m \in \mathbf{M}}} P(y_{a_i,z,m}, a_z, z, m_{a_i,z}) \Big) \tag{28}$$

$$= \frac{1}{P(a_j)} \sum_{\substack{z \in \mathbf{Z}, \\ m \in \mathbf{M}}} P(y|z,m,do(a_i)) P(m \mid do(a_j), z) P(z) \tag{29}$$

$$- \frac{P(a_i)}{P(a_j)} \sum_{\substack{z \in \mathbf{Z}, \\ m \in \mathbf{M}}} P(y \mid m, z, a_i) P(m, z \mid a_j)$$

$$- \frac{1}{P(a_j)} \sum_{\substack{z \in \mathbf{Z}, \\ m \in \mathbf{M}}} (P(y, z, m \mid do(a_i)) - P(y, z, m \mid a_i) P(a_i)),$$

$$\text{SE}_{a_i,a_j}(y) = P(y_{a_i} \mid a_j) - P(y \mid a_i) \tag{30}$$

$$= \frac{1}{P(a_j)} \sum_{\substack{z \in \mathbf{Z}, \\ m \in \mathbf{M}}} P(y_{a_i,z,m}, a_z, z, m_{a_i,z}) - P(y \mid a_i) \tag{31}$$

$$= \frac{1}{P(a_j)} \sum_{\substack{z \in \mathbf{Z}, \\ m \in \mathbf{M}}} (P(y, z, m \mid do(a_i)) - P(y, z, m \mid a_i) P(a_i)) - P(y \mid a_i).$$

In the above equations, the effects with do-interventions are unidentifiable in the presence of unobserved confounding. Furthermore, we present details on the performed calculations in the proofs of Lemma 2 and Lemma 3.

### D.3 COUNTERFACTUAL IDENTIFICATION AND BOUNDS

We use the following corollary.

**Corollary 1.** *Let $X$ and $A$ be two variables of a structural causal model. By consistency it holds*

$$P(X_a) = P(X \mid A = a)P(A = a) + P(X_a \mid A \neq a)P(A \neq a). \tag{32}$$

*Proof.* Follows directly from basic probability theory. □

We start by deriving bounds for expressions of the form $P(y_{a_i} \mid a)$. If $a = a_i$, there is nothing to show, since, by consistency, we have $P(y_{a_i} \mid a_i) = P(y \mid a_i)$. Therefore, let $a \neq a_i$ in the following.

**Lemma 2.** *For a binary sensitive attribute with $a_i \neq a_j$, it holds*

$$P^+(y_{a_i} \mid a_j) = \frac{1}{P(a_j)} \sum_{\substack{z \in \mathbf{Z}, \\ m \in \mathbf{M}}} P^+(y \mid z, m, a_i) P^+(m \mid a_i, z) P(z) - \frac{P(a_i)}{P(a_j)} \sum_{\substack{z \in \mathbf{Z}, \\ m \in \mathbf{M}}} P(y, z, m \mid a_i),$$
$$\tag{33}$$

$$P^-(y_{a_i} \mid a_j) = \frac{1}{P(a_j)} \sum_{\substack{z \in \mathbf{Z}, \\ m \in \mathbf{M}}} P^-(y \mid z, m, a_i) P^-(m \mid a_i, z) P(z) - \frac{P(a_i)}{P(a_j)} \sum_{\substack{z \in \mathbf{Z}, \\ m \in \mathbf{M}}} P(y, z, m \mid a_i).$$
$$\tag{34}$$

*Proof.* By Theorem 3, we can write

$$
\begin{aligned}
P(y_{a_i} \mid a_j) &= \sum_{\substack{z \in \mathbf{Z}, \\ m \in \mathbf{M}}} P(y_{a_i}, z, m_{a_i} \mid a_j) \\
&= \frac{1}{P(a_j)} \sum_{\substack{z \in \mathbf{Z}, \\ m \in \mathbf{M}}} \Big[ P(y, z, m \mid \mathrm{do}(a_i)) - P(y, z, m \mid a_i) P(a_i) \\
&\quad - \sum_{\substack{a' \neq a_i, \\ a' \neq a_j}} P(y_{a_i}, z, m_{a_i} \mid a') P(a') \Big].
\end{aligned}
\tag{35}
$$

For binary sensitive attributes, this reduces to

$$
P(y_{a_i} \mid a_j) = \frac{1}{P(a_j)} \sum_{\substack{z \in \mathbf{Z}, \\ m \in \mathbf{M}}} [P(y, z, m \mid \mathrm{do}(a_i)) - P(y, z, m \mid a_i) P(a_i)].
\tag{36}
$$

The first term is not identifiable but can be bounded in a GMSM by rewriting the term as

$$
\sum_{\substack{z \in \mathbf{Z}, \\ m \in \mathbf{M}}} P(y, z, m \mid \mathrm{do}(a_i)) = \sum_{\substack{z \in \mathbf{Z}, \\ m \in \mathbf{M}}} P(y \mid z, m, \mathrm{do}(a_1)) P(m \mid \mathrm{do}(a_1), z) P(z).
\tag{37}
$$

Then, the desired statement follows by Theorem 2. $\qquad\square$

With the results from above, we turn to the derivation of bounds for the second factor $P(y_{a_i, m_{a_j}} \mid a)$. Based on the causal explanation formula, we only have to consider the case $a = a_j$.

**Lemma 3.** *For binary sensitive attributes, it holds*

$$
\begin{aligned}
P^+(y_{a_i, m_{a_j}} \mid a_j) &= \frac{1}{P(a_j)} \sum_{\substack{z \in \mathbf{Z}, \\ m \in \mathbf{M}}} P^+(y \mid z, m, a_i) P^+(m \mid a_j, z) P(z) \\
&\quad - \frac{P(a_i)}{P(a_j)} \sum_{\substack{z \in \mathbf{Z}, \\ m \in \mathbf{M}}} P(y | m, z, a_i) P(m \mid z, a_j) P(z),
\end{aligned}
\tag{38}
$$

$$
\begin{aligned}
P^-(y_{a_i, m_{a_j}} \mid a_j) &= \frac{1}{P(a_j)} \sum_{\substack{z \in \mathbf{Z}, \\ m \in \mathbf{M}}} P^-(y \mid z, m, a_i) P^-(m \mid a_j, z) P(z) \\
&\quad - \frac{P(a_i)}{P(a_j)} \sum_{\substack{z \in \mathbf{Z}, \\ m \in \mathbf{M}}} P(y \mid m, z, a_i) P(m \mid z, a_j) P(z).
\end{aligned}
\tag{39}
$$

*Proof.* According to Section D.2, we can write

$$
\begin{aligned}
P(y_{a_i, m_{a_j}} \mid a_j) &= \frac{1}{P(a_j)} \sum_{\substack{z \in \mathbf{Z}, \\ m \in \mathbf{M}}} P(y_m \mid m_{a_j}, z, do(a_i)) P(m \mid z, do(a_j)) P(z) \\
&\quad - \frac{P(a_i)}{P(a_j)} \sum_{\substack{z \in \mathbf{Z}, \\ m \in \mathbf{M}}} P(y \mid m, z, a_i) P(m_{a_j}, z \mid a_i) \\
&\quad - \frac{1}{P(a_j)} \sum_{\substack{z \in \mathbf{Z}, \\ m \in \mathbf{M}}} \sum_{l \neq i, j} P(y_{a_i, m}, m_{a_j}, z \mid a_l) P(a_l).
\end{aligned}
\tag{40}
$$

If $A$ is binary, this reduces to

$$
\begin{aligned}
P(y_{a_i,m_{a_j}} \mid a_j) = {} & \frac{1}{P(a_j)} \sum_{\substack{z \in \mathbf{Z}, \\ m \in \mathbf{M}}} P(y \mid z, m, do(a_i)) P(m \mid do(a_j), z) P(z) \\
& + \sum_{\substack{z \in \mathbf{Z}, \\ m \in \mathbf{M}}} P(y \mid m, z, a_i) P(m, z \mid a_j) - \frac{1}{P(a_j)} \sum_{\substack{z \in \mathbf{Z}, \\ m \in \mathbf{M}}} P(y_m, m_{a_j}, z \mid a_i).
\end{aligned}
\tag{41}
$$

We further use

$$
\begin{aligned}
P(y_{a_i,m}, m_{a_j}, z \mid a_i) = {} & \frac{1}{P(a_i)} P(y \mid m_{a_j}, z, do(a_i)) P(m \mid z, do(a_j)) P(z) \\
& - \frac{P(a_j)}{P(a_i)} P(y_m, m_{a_j}, z \mid do(a_i), a_j),
\end{aligned}
\tag{42}
$$

and, then, the overall expression follows, i.e.,

$$
\begin{aligned}
P(y_{a_i,m_{a_j}} \mid a_j) = {} & \frac{1}{P(a_j)} \sum_{\substack{z \in \mathbf{Z}, \\ m \in \mathbf{M}}} P(y|z, m, do(a_i)) P(m \mid do(a_j), z) P(z) \\
& - \frac{P(a_i)}{P(a_j)} \sum_{\substack{z \in \mathbf{Z}, \\ m \in \mathbf{M}}} P(y \mid m, z, a_i) P(m, z \mid a_j).
\end{aligned}
\tag{43}
$$

$\square$

Overall, we want to bound the path-specific causal fairness effects (i.e., Ctf-DE, Ctf-IE, and Ctf-SE), which consist of differences of the two expressions analyzed above. In the following, we derive the bounds for each counterfactual factor.

**Proof for Theorem 1:**

Employing Theorem 2, Lemma 2, and Lemma 3, the counterfactual direct effect is upper bounded by

$$
\mathrm{DE}^{+}_{a_i,a_j}(y \mid a_i) = P^{+}(y_{a_j,m_{a_i}} \mid a_i) - P^{-}(y_{a_i} \mid a_i)
\tag{44}
$$

$$
= \frac{1}{P(a_i)} \sum_{\substack{z \in \mathbf{Z}, \\ m \in \mathbf{M}}} P^{+}(y \mid z, m, a_j) P^{+}(m \mid a_i, z) P(z)
\tag{45}
$$

$$
- \frac{P(a_j)}{P(a_i)} \sum_{\substack{z \in \mathbf{Z}, \\ m \in \mathbf{M}}} P(y \mid m, z, a_j) P(m \mid z, a_i) P(z) - P(y \mid a_i)
$$

and lower bounded by

$$
\mathrm{DE}^{-}_{a_i,a_j}(y \mid a_i) = P^{-}(y_{a_j,m_{a_i}} \mid a_i) - P^{+}(y_{a_i} \mid a_i)
\tag{46}
$$

$$
= \frac{1}{P(a_i)} \sum_{\substack{z \in \mathbf{Z}, \\ m \in \mathbf{M}}} P^{-}(y \mid m, z, a_j) P^{-}(m \mid z, a_i) P(z)
\tag{47}
$$

$$
- \frac{P(a_j)}{P(a_i)} \sum_{\substack{z \in \mathbf{Z}, \\ m \in \mathbf{M}}} P(y \mid m, z, a_j) P(m \mid z, a_i) P(z) - P(y \mid a_i).
$$

The results for the indirect counterfactual effect follow directly by setting

$$
\mathrm{IE}^{+}_{a_i,a_j}(y \mid a_j) = P^{+}(y_{a_i,m_{a_j}} \mid a_j) - P^{-}(y_{a_i} \mid a_j)
\tag{48}
$$

and

$$
\mathrm{IE}^{-}_{a_i,a_j}(y \mid a_j) = P^{-}(y_{a_i,m_{a_j}} \mid a_j) - P^{+}(y_{a_i} \mid a_j).
\tag{49}
$$

Similarly, the bounds for the spurious counterfactual effect can be derived to

$$
\mathrm{SE}^{+}_{a_i,a_j}(y) = P^{+}(y_{a_i} \mid a_j) - P(y \mid a_i)
\tag{50}
$$

and

$$
\mathrm{SE}^{-}_{a_i,a_j}(y) = P^{-}(y_{a_i} \mid a_j) - P(y \mid a_i).
\tag{51}
$$

# E EXTENDED THEORY

## E.1 COMPATIBILITY OF SCMs

An acyclic SCM $\mathcal{C} = (\mathbf{V}, \mathbf{U}_{\mathcal{C}}, \mathcal{F}, P_{\mathbf{U}})$, such that $\mathbf{U} \subseteq \mathbf{U}_{\mathcal{C}}$, $\mathcal{G} \subseteq \mathcal{G}_{\mathcal{C}}$, is *compatible* with sensitivity model $S$ if $\mathbf{U}_{\mathcal{C}}$ does not introduce additional unobserved confounding and the probability distribution $P_{\mathbf{V} \cup \mathbf{U}_c}$ induced by $\mathcal{C}$ belongs to $\mathcal{P}$, i.e., $P_{\mathbf{V} \cup \mathbf{U}_c} \in \mathcal{P}$.

We further assume no additional unobserved confounding. That is, we have: For each $M$ denote $Pa_{\mathcal{G}}(M)$ the observed parents in $\mathbf{V}$. Then, $\mathbf{U}$ is a valid adjustment set for the relationship between $Pa_{\mathcal{G}}(M)$ and $M$, i.e., $P(m \mid do(Pa_{\mathcal{G}}(M) = pa)) = \int P(m \mid do(Pa_{\mathcal{G}}(M) = pa), \mathbf{U} = \mathbf{u}) P(\mathbf{U} = \mathbf{u}) d\mathbf{u}$.

## E.2 EXTENSION OF THEOREM 1

In the following, we provide an extension of Theorem 1 for bounding the counterfactual effects of continuous outcome variables.

**Lemma 4.** *Let $F(y)$ denote the CDF of $P(y \mid m, z, a_i)$ for a continuous outcome $Y$. Then, the probability functions $p^+(y \mid m, z, a_i)$ and $p^-(y \mid m, z, a_i)$ are given by*

$$
p^+(y \mid m, z, a_i) = \begin{cases} ((1 - \Gamma_Y)^{-1} P(a_i \mid z) + \Gamma_Y^{-1}) p(y \mid m, z, a_i), & \text{if } F(y) \leq \frac{\Gamma_Y}{1 + \Gamma_Y} \\ ((1 - \Gamma_Y) P(a_i \mid z) + \Gamma_Y) p(y \mid m, z, a_i). & \text{if } F(y) > \frac{\Gamma_Y}{1 + \Gamma_Y}. \end{cases} \tag{52}
$$

*and*

$$
p^-(y \mid m, z, a_i) = \begin{cases} ((1 - \Gamma_Y) P(a_i \mid z) + \Gamma_Y) p(y \mid m, z, a_i), & \text{if } F(y) \leq \frac{1}{1 + \Gamma_Y} \\ ((1 - \Gamma_Y)^{-1} P(a_i \mid z) + \Gamma_Y^{-1}) p(y \mid m, z, a_i), & \text{if } F(y) > \frac{1}{1 + \Gamma_Y}. \end{cases} \tag{53}
$$

The bounds require estimating the observational density $p(y \mid z, m, a)$. Nevertheless, in practice, we can only observe the empirical distribution from which we can obtain $\hat{p}(y \mid z, m, a)$ via an arbitrary conditional density estimator. The work in (Frauen et al., 2023b) introduces importance sampling estimators for finite sample bounds, which can be adapted to estimate the observational density.

## E.3 COUNTERFACTUAL IDENTIFIABILITY FOR SUB-PROBLEMS

Our work obtains bounds on the counterfactual effects in a general setting with unobserved confounding, i.e., $\mathbf{U} = \{U_{\text{DE}}, U_{\text{IE}}, U_{\text{SE}}\} \neq 0$. Now, we state the counterfactual identification results for subproblems in which subsets of $\mathbf{U}$ are considered to be zero.

In Section D, we employ counterfactual unnesting theory to split the expressions $P(y_{a_i} \mid a)$ and $P(y_{a_i, m_{a_j}} \mid a)$ into the respective counterfactual factors. For the first factor, we derive

$$
P(y_{a_0} \mid a) = \frac{1}{P(a)} \sum_{z, m} P(y_{a_0, z, m}, a_z, z, m_{a_0, z}). \tag{54}
$$

If we assume that there exists unobserved confounding between sensitive attribute and mediator ($U_{\text{IE}}$), sensitive attribute and covariates ($U_{\text{SE}}$), and sensitive attribute and outcome ($U_{\text{DE}}$), this expression cannot be reduced further. Other possible scenarios of unobserved confounding are exam-

ined in the following. By the counterfactual factorization theorem, we yield

$$U_{\text{DE}} = U_{\text{IE}} = U_{\text{SE}} = 0: \qquad P(y_{a_0}, a) = \sum_{z,m} P(y_{a_0,z,m}) P(a_z) P(z) P(m_{a_0,z}), \qquad (55)$$

$$U_{\text{SE}} \neq 0,\ U_{\text{DE}} = U_{\text{IE}} = 0: \qquad P(y_{a_0}, a) = \sum_{z,m} P(y_{a_0,z,m}) P(a_z, z) P(m_{a_0,z}), \qquad (56)$$

$$U_{\text{IE}} \neq 0,\ U_{\text{DE}} = U_{\text{SE}} = 0: \qquad P(y_{a_0}, a) = \sum_{z,m} P(y_{a_0,z,m}) P(a_z, m_{a_0,z}) P(z), \qquad (57)$$

$$U_{\text{DE}} \neq 0,\ U_{\text{SE}} = U_{\text{IE}} = 0: \qquad P(y_{a_0}, a) = \sum_{z,m} P(y_{a_0,z,m}, a_z) P(z) P(m_{a_0,z}), \qquad (58)$$

$$U_{\text{SE}}, U_{\text{IE}} \neq 0,\ U_{\text{DE}} = 0: \qquad P(y_{a_0}, a) = \sum_{z,m} P(y_{a_0,z,m}) P(a_z, z, m_{a_0,z}), \qquad (59)$$

$$U_{\text{SE}}, U_{\text{DE}} \neq 0,\ U_{\text{IE}} = 0: \qquad P(y_{a_0}, a) = \sum_{z,m} P(y_{a_0,z,m}, a_z, z) P(m_{a_0,z}), \qquad (60)$$

$$U_{\text{IE}}, U_{\text{DE}} \neq 0,\ U_{\text{SE}} = 0: \qquad P(y_{a_0}, a) = \sum_{z,m} P(y_{a_0,z,m}, a_z, m_{a_0,z}) P(z). \qquad (61)$$

All terms in the first two cases are directly identifiable from the data. In all other cases, one term is unidentifiable for $a \neq a_0$. To derive our bounds, we focus on the general case, allowing for all three types of unobserved confounding. The expression for the second term of interest follows similarly.

### E.4 Training algorithm for fair prediction

In the following, we discuss Algorithm 1 in more detail. Depending on the task, i.e., binary classification, multi-class classification, or regression, the precise calculations of the constraints might vary. Therefore, we will discuss all tasks separately:

1. **Binary classification:** Assume $A, Y \in \{0, 1\}$. For each of the three different fairness effects $\text{CF} \in \{\text{DE}, \text{IE}, \text{SE}\}$, we are interested in, e.g., $\text{CF}(Y = 1 \mid A = 1)$. Due to symmetry reasons for binary sensitive attributes $A$ (e.g., discrimination of women compared to men is symmetric to discrimination of men compared to women), it is sufficient to focus on one realization of the sensitive attribute. Therefore, we obtain only one constraint per $\text{CF} \in \{\text{DE}, \text{IE}, \text{SE}\}$, i.e., three constraints in total.

2. **Regression:** For continuous outcome variables $Y \in \mathbb{R}$, it is reasonable to consider the expectation over $Y$. Therefore, for this task, we also obtain one constraint per CF and thus three constraints overall.

3. **Multi-class classification:** For this task, two different options exist for defining the constraints we aim to optimize over. Option one follows the same reasoning as in binary classification and regression in that one can constrain the expected CF over all realizations of $Y$ for each $\text{CF} \in \{\text{DE}, \text{IE}, \text{SE}\}$. A more fine-grained approach is to constrain each CF for each possible realization of $Y$, $y \in \mathcal{Y}$. As a result, we need to optimize our predictor with respect to $3 \times |\mathcal{Y}|$ constraints. Although this is feasible for binary classification, it can be highly challenging for large $|\mathcal{Y}|$. Which of these options to choose for a specific problem is left to the practitioner.

Algorithm 1 (main paper) provides the training procedure for binary classification and regression. In Algorithm 2, we present an alternative training procedure for multi-class classification in which we impose one constraint for each $\text{CF} \in \{\text{DE}, \text{IE}, \text{SE}\} \times y \in \mathcal{Y}$.

---

**Algorithm 2:** Alternative training procedure for fair multi-class classification models robust to unobserved confounding

---

**Input:** Data $\{(A_i, Z_i, M_i, Y_i) : i \in \{1, \ldots, n\}\}$, sensitivity parameter $\Gamma_M$, vector of fairness constraints $\boldsymbol{\gamma}$, Lagrangian parameter vectors $\boldsymbol{\lambda}_0$ and $\boldsymbol{\mu}_0$, pre-trained density estimators $g_A, g_M$, initial prediction model $f_{\theta_0}$, convergence criterion $\varepsilon$, Lagrangian update-rate $\alpha$

**Output:** Fair predictions $\{\hat{y}_i : i \in \{1, \ldots, n\}\}$, trained prediction model $f_\theta^*$

```
        /* Initialize parameters                                                          */
1   f_{θ_j} ← f_{θ_0} ; λ_k ← λ_0 ; μ_k ← μ_0 for k ∈ max iterations do
            /* Train prediction model                                                     */
2       for l ∈ nested epochs do
3           ŷ ← f_{θ_l}(A, Z, M);
                /* Determine CF⁺, CF⁻ for CF ∈ {DE, IE, SE} following Theorem 1 for each y ∈ 𝒴    */
4           CF⁺_y ← ub(f_{θ_l}, (A, Z, M), Γ_M, g_A, g_M, y),   ∀y ∈ 𝒴;
5           CF⁻_y ← lb(f_{θ_l}, (A, Z, M), Γ_M, g_A, g_M, y),   ∀y ∈ 𝒴;
                /* Optimization objective following Eq. (11)                              */
6           for CF ∈ {DE, IE, SE}, ∀y ∈ 𝒴 do
7               c_{CF_y} ← max{|CF⁺_y|, |CF_y⁻|}
8           end
9           c ← [c_{DE_{y_1}}, . . . , c_{DE_{y_d}}, c_{IE_{y_1}}, . . . , c_{IE_{y_d}}, c_{SE_{y_1}}, . . . , c_{SE_{y_d}}]^T,   where d = |𝒴|;
10          lagrangian ← loss(f_{θ_l}) − λ_k(γ − c) − (1/2μ_k)(λ_k − λ_{k−1})²;
                /* Update parameters of predictor                                         */
11          f_{θ_{l+1}} ← optimizer(lagrangian, f_{θ_l})
12      end
13      if c ≤ γ then
                /* Check for convergence                                                  */
14          if prediction loss(f_{θ_l}) ≤ ε then
15              f_θ^* ← f_{θ_{l+1}};
16              stop
17          end
18      end
            /* Update Lagrangian parameters                                               */
19      λ_{k+1} ← max{λ_k − cμ_k, 0 } ; μ_{k+1} ← αμ_k
20  end
```

---

We note that, depending on the task, the algorithms optimize with respect to three or $3 \times |\mathcal{Y}|$ constraints. Therefore, the vectors $\boldsymbol{\lambda}, \boldsymbol{\mu}, \boldsymbol{\gamma}$ and the final fairness vector $\mathbf{c}$ are also of these dimensions.

We also note the following: In Algorithm 1 (main paper), the maximum in Line 7 is always taken over the two values $|\mathbb{E}(\mathrm{CF}^+)|$ and $|\mathbb{E}(\mathrm{CF}^-)|$ for each $\mathrm{CF} \in \{\mathrm{DE}, \mathrm{IE}, \mathrm{SE}\}$. In Algorithm 2, the maximum in Line 7 is computed over the two values $|\mathrm{CF}_y^+|$ and $|\mathrm{CF}_y{}^-|$ for each $\mathrm{CF} \times y \in |\mathcal{Y}|$. The two algorithms thus differ concerning the number of evaluations of the maximum, i.e.. three evaluations in Alg. 1 (main paper) and $3 \times |\mathcal{Y}|$ in Alg. 2.

# F  BOUNDS ON FURTHER FAIRNESS METRICS

Here, we show that our framework is general and can be applied to a broad set of fairness notions. Our main paper focused on fairness bounds based on the path-specific causal fairness effects (Zhang & Bareinboim, 2018a). Nevertheless, our approach is general and can be easily applied to other notions of fairness as well. In the following, we outline the derivation of bounds for the fair on average causal effect (Khademi et al., 2019) and path-specific individual fairness (Chiappa, 2019; Wu et al., 2019b).

## F.1  EXTENSION TO THE FAIR ON AVERAGE CAUSAL EFFECT (FACE)

Khademi et al. (2019) introduced the definition of a *fair on average causal effect (FACE)* of group fairness, which we denote by $\text{FACE}(a_i)$. We further define the generalized form of the effect for non-binary attributes as *average FACE*

$$\text{AFACE}(\mathbf{A}) := \mathbb{E}_{\mathbf{A}}[\text{FACE}(a_i)] = \mathbb{E}_{\mathbf{A}}[\mathbb{E}[Y \mid \text{do}(A = a_i)]] - \mathbb{E}[Y \mid \text{do}(A = a_0)]. \tag{62}$$

A fair predictor should ideally return $\text{FACE}(a_i) = 0$ for all $a_i \in \mathbf{A}$, which implies $\text{AFACE}(\mathbf{A}) = 0$. Nevertheless, the single effects might not always be identifiable. In these cases, the average FACE can be used as a relaxation.

For $j = 1, \ldots, l$, $k = |\mathbf{A}|$, we define

$$\mathbb{E}[\hat{Y}] = \sum_{z \in \mathbf{Z}} \sum_{m \in \mathbf{M}} \mathbb{E}[\hat{Y} \mid z, m, \text{do}(a_j)] P(m \mid z, \text{do}(a_j)) P(z).$$

The upper and lower bounds for $\mathbb{E}[\hat{Y} \mid z, m, \text{do}(a_j)]$ and $P(m \mid z, \text{do}(a_j))$ can iteratively be derived through the algorithm for causal sensitivity analysis with mediators presented in (Frauen et al., 2023b). The upper bound $ub_{a_j}$ and lower bound $lb_{a_j}$ of $\text{FACE}(a_j)$ (with respect to baseline $a_0$) are then given by

$$ub_{a_j} = \sum_{z \in \mathbf{Z}} P(z)[\sum_{m \in \mathbf{M}} (\mathbb{E}^+(Y \mid z, m, a_j) P^+(m \mid z, a_j) - \mathbb{E}^-(Y \mid z, m, a_0) P^-(m \mid z, a_j))], \tag{63}$$

$$lb_{a_j} = \sum_{z \in \mathbf{Z}} P(z)[\sum_{m \in \mathbf{M}} (\mathbb{E}^-(Y \mid z, m, a_0) P^-(m \mid z, a_j)) - \mathbb{E}^+(Y \mid z, ma_j) P^+(m \mid z, a_j)]. \tag{64}$$

For the bounds for $\text{AFACE}(\mathbf{A})$, $|\mathbf{A}| = k$, it follows that

$$\text{ub} = \frac{1}{k-1} \sum_{a' \in \mathbf{A} \setminus \mathbf{a_0}} ub_{a'}, \qquad \text{lb} = \frac{1}{k-1} \sum_{a' \in \mathbf{A} \setminus \mathbf{a_0}} lb_{a'}. \tag{65}$$

We note that the above bounds should be used cautiously since highly positive and negative effects can cancel out in the definition for AFACE. However, this is not due to our derivation but due to the definition of considering average effects at the population level without analyzing the heterogeneity across the population.

## F.2  EXTENSION TO PATH-SPECIFIC INDIVIDUAL FAIRNESS

Path-specific individual fairness (Chiappa, 2019; Wu et al., 2019b) along the path $\pi$ wrt. sensitive attributes $a_0, a_1$ is achieved, if for all $z \in Z$,

$$\mathbb{E}_{Y_{a_0}, Y_{a_1}}[Y_{a_1 \mid \pi} - Y_{a_0} \mid Z = z] = 0 \tag{66}$$

holds. To build a fair prediction framework based on this fairness notion, we need to derive bounds on the expressions $P(Y_{a_1 \mid \pi} \mid Z = z)$ and $P(Y_{a_0} \mid Z = z)$.

For simplicity, we assume wlog. that $\pi$ represents the path $A \to M \to Y$ for a single mediator $M$. Then, for the first term, it holds that

$$P(Y_{a_1 | \pi} = y \mid Z = z) \tag{67}$$

$$= \sum_{m \in \mathcal{M}} P(Y_{a_1} = y, M = m_{a_1} \mid Z = z) \tag{68}$$

$$= \sum_{m \in \mathcal{M}} P(Y = y \mid do(A = a_1), M = m, Z = z) P(M = m \mid do(A = a_1), Z = z). \tag{69}$$

The second term directly decomposes into

$$P(Y_{a_0} = y \mid Z = z) \tag{70}$$

$$= \sum_{m \in \mathcal{M}} P(Y = y \mid do(A = a_0), M = m, Z = z) P(M = m \mid do(A = a_0), Z = z). \tag{71}$$

We observe that both terms constitute subproblems of our bounds derived in Supplement D. The terms do not contain nested counterfactual expressions but only interventional effects. Therefore, we can obtain the bounds from Theorem 2.

## G  IMPLEMENTATION DETAILS

We implemented the experiments in PyTorch Lightning. Code and data for the reproducibility of the results are provided in our GitHub repository.

All neural network classifiers used in our study consisted of three layers with leaky ReLU activation function and dropout. For optimization, we used Adam (Kingma & Ba, 2015).

Our proposed fair predictor training framework requires a pre-trained model for estimating the density of the mediators based on the confounders and the sensitive attribute. The density of the sensitive attribute conditioned on the confounders is taken as the relative frequency. Since we only consider discrete mediators, a simple multi-class classifier is sufficient to perform the density estimation. We use the same neural network architecture for mediator prediction as for our standard classifier.

Hyperparameters of the classification models were kept fixed across the three classifiers on the simulated datasets for better comparability. The models consisted of one hidden layer of size ten and a dropout layer with a rate of $0.1$ and were trained with a batch size of 128 and an initial learning rate of 0.0001. We trained the fair naïve model and the fair robust model with a fairness constraint of $\gamma = 0.02$, initial Lagrangian parameters $\lambda = 0.1$, $\mu = 0.02$, and an update rate of $\alpha = 1.5$. The prediction performance presented in terms of the ROC AUC is presented in Table 4. To facilitate the evaluation of the overall performance (i.e. performance and fairness performance), we introduce a *fairness utility* function.

**Definition 3** (Fairness utility). We define the utility of a fair predictor as the weighted sum of its fairness and its prediction performance measured as the ROC AUC: For $\omega \in [0, 1]$

$$\mathcal{U}(f_\theta) = \omega R(f_\theta) - (1 - \omega)F(f_\theta), \tag{72}$$

where $F$ represents the fairness evaluated through a causal fairness notion and $R$ the ROC AUC.

We set $F(f_\theta) = \frac{1}{3}(\max\{|\mathrm{DE}^+_\mathbb{E}|, |\mathrm{DE}^-_\mathbb{E}|\} + \max\{|\mathrm{IE}^+_\mathbb{E}|, |\mathrm{IE}^-_\mathbb{E}|\} + \max\{|\mathrm{SE}^+_\mathbb{E}|, |\mathrm{SE}^-_\mathbb{E}|\})$ and $\omega = 0.5$ for our calculations. We report the performance in terms of the fairness utilities of all classifiers as well in Table 4. Note that the defined utility is upper bounded by one but does not have a lower bound. Thus, negative utilities are possible.

Table 4: Performance of the prediction models on the simulated datasets measured by the ROC AUC and the fairness utility.

| Experiment | ROC AUC | | | Fairness utility | | |
|---|---|---|---|---|---|---|
| | Standard | Fair naïve | Fair robust | Standard | Fair naïve | Fair robust |
| $\Phi_{U_{\mathrm{DE}}} = 1$ | 0.8579 | 0.8112 | 0.6780 | $-0.7284$ | $-0.8074$ | $-0.5004$ |
| $\Phi_{U_{\mathrm{DE}}} = 2$ | 0.8245 | 0.7885 | 0.7618 | $-0.7402$ | $-0.8138$ | $-0.4662$ |
| $\Phi_{U_{\mathrm{DE}}} = 3$ | 0.8037 | 0.8003 | 0.5226 | $-0.7550$ | $-0.8137$ | $-0.5871$ |
| $\Phi_{U_{\mathrm{DE}}} = 4$ | 0.7535 | 0.7466 | 0.5754 | $-0.7809$ | $-0.8409$ | $-0.5741$ |
| $\Phi_{U_{\mathrm{IE}}} = 1$ | 0.8929 | 0.8671 | 0.7187 | $-0.9026$ | $-0.6405$ | $-0.5334$ |
| $\Phi_{U_{\mathrm{IE}}} = 2$ | 0.9150 | 0.8667 | 0.6856 | $-0.8948$ | $-0.6348$ | $-0.5332$ |
| $\Phi_{U_{\mathrm{IE}}} = 3$ | 0.8964 | 0.8271 | 0.7382 | $-0.9066$ | $-0.6497$ | $-0.5145$ |
| $\Phi_{U_{\mathrm{IE}}} = 4$ | 0.8557 | 0.8254 | 0.7321 | $-0.9276$ | $-0.6581$ | $-0.5290$ |

For our real-world case study, we also employed three-layered networks with leaky ReLU activation function and dropout. The robustly fair model was trained with a fairness constraint of $\gamma = 0$, initial Lagrangian parameters $\lambda = 3.0$, $\mu = 1.5$, an update rate of $\alpha = 1.5$, and a sensitivity parameter of $\Gamma = 2$. The hyperparameters (hidden layer dimension, dropout rate, learning rate) of the density estimator and prediction model were optimized with the library Optuna (`https://optuna.org/`) across 100 trials. The final parameters and the prediction performance measured by the MSE are stated in Table 5.

Table 5: Hyperparameters and prediction performance of the mediator density estimator and prediction model on the real-world dataset.

| Model | MSE | Learning rate | Hidden dimension | Dropout |
|---|---|---|---|---|
| Density estimator | — | 0.0009 | 256 | 0.2100 |
| Standard predictor | 3.2383 | 0.0294 | 32 | 0.3175 |
| Robustly fair predictor | 4.9607 | 0.0001 | 64 | 0.4503 |

## H  EVALUATION SETTINGS

### H.1  SYNTHETIC DATASET

We consider two settings with different types of unobserved confounding. Both contain a single binary sensitive attribute, mediator, confounder, and outcome variable. We call the first setting "$U_{\text{DE}}$". Therein, we introduce an unobserved confounder on the direct effect with confounding level $\Phi$. We call the second setting "$U_{\text{IE}}$". Therein, we introduce an unobserved confounder on the indirect effect with confounding level $\Phi$.

We then generate synthetic datasets for each setting and confounding level. Specifically, we draw each 20,000 samples from the following structural equations for $U_{\text{DE}}, U_{\text{IE}} \sim \mathcal{N}(\Phi, e^{-4})$ with $\Phi \in [1, 4]$:

$$
\begin{aligned}
Z &\sim \text{Bernoulli}(0.5) \\
A &\sim \text{Bernoulli}(\sigma(5Z - U_{\text{DE}})) \\
M &\sim \text{Bernoulli}(\sigma(4A + 2Z) \\
Y &\sim \text{Bernoulli}(\sigma(3A + Z + 2M - U_{\text{DE}})),
\end{aligned}
$$

and

$$
\begin{aligned}
Z &\sim \text{Bernoulli}(0.5) \\
A &\sim \text{Bernoulli}(\sigma(5Z - U_{\text{IE}})) \\
M &\sim \text{Bernoulli}(\sigma(4A + 2Z - U_{\text{IE}}) \\
Y &\sim \text{Bernoulli}(\sigma(3A + Z + 2M)),
\end{aligned}
$$

where $\sigma$ is the sigmoid function. We restrict the Bernoulli probability to fall in the interval $[0.02, 0.98]$ to guarantee overlap. The simulation is performed through the partially randomized causal simulator PARCS (`https://pypi.org/project/pyparcs/`).

### H.2  REAL-WORLD DATASET

Our real-world study is based on the U.S. Survey of Prison Inmates (SPI); see United States. Bureau of Justice Statistics (2021). Based on the survey data, we created a dataset for predicting prison sentences for drug offenders. We consider the race of the defendant as the sensitive attribute and prison history as a mediating factor. Overall, the prison sentence should only depend on the type of offense. As unobserved confounders, we consider the defendant's family's crime history and citizenship. Formally, we filter the survey data based on the following criteria:

- General filter criterion "drug offense": Our analysis only considers prisoners sentenced for drug offenses. Therefore, we filter the complete survey data for offense type "drug"'. Furthermore, we split the type into "trafficking" and "possession", represented through a binary variable, and disregard other non-specified drug offenses.

- Race: We aim to diminish the effects of race, especially "white American", on the prison sentence. Therefore, we combine all information about the defendant's race into a binary variable representing "white" and "non-white," respectively.

- Offense type: We encode the offense type into multiple binary variables indicating drug possession, manufacturing or growing, import of drugs to the US, distribution of drugs, and drug money laundry.

- Sentence: We use information about the prison sentence length in months as a continuous target variable. We restrict the imprisonment length to a maximum of one year for our analysis.

- Crime history: Prior court verdicts can characterize crime history. We encode information about the defendant's prior sentences into a discrete variable indicating the severity of the crime history, if any.

- Family history: We are interested if a family member of the defendant had been sentenced to prison before. Therefore, we combine all survey information about the former imprisonment of parents, children, and spouses into a binary variable indicating if at least one family member had been imprisoned before.

- Citizenship / Immigrant to the U.S.: The defendant's citizenship (if it is non-U.S.) is not included in the publicly available data due to privacy concerns. Hence, we use information about U.S. citizenship as a proxy, which we encode as a binary variable.

Individuals who did not provide an answer to the respective question or who answered "I do not know" were excluded from our analysis. We imputed missing target values through $k$NN-imputation with 10 neighbors. We split the resulting dataset in the ratio 60%, 20% , 20% for training, validating, and testing our models, respectively.

# I  ADDITIONAL EXPERIMENTAL RESULTS

In the following, we present further experimental results to demonstrate our framework.

## I.1  BOUNDS FOR VARYING SENSITIVITY PARAMETERS

In Figure 10 and Figure 11, we present results for the fairness of the predictions. Here, we examine the three classifiers as in our main paper but when using varying sensitivity parameters variable in $\{1.2, 2.0, 5.0\}$ for calculating the bounds. We observe the same behavior as in our main paper: the interval defined through the bounds increases in the sensitivity parameter. For the standard prediction model, the interval eventually covers the original path-specific effect in the data. Together, the results further demonstrate the effectiveness of our framework.

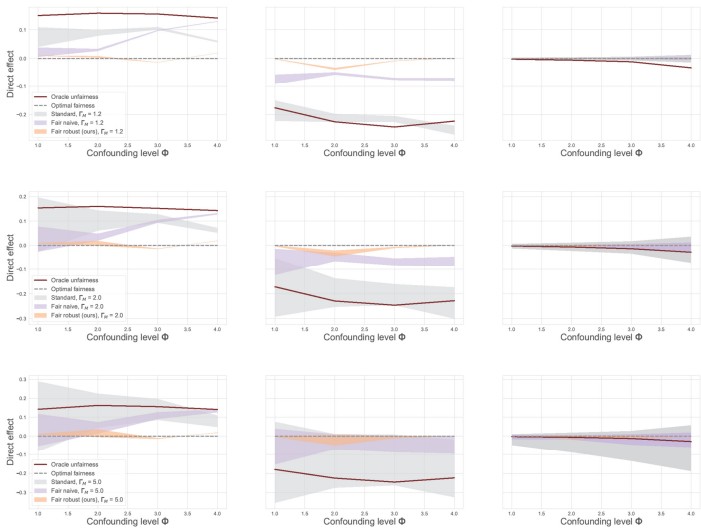

Figure 10: Bounds of the three prediction models on the dataset including *direct unobserved confounding* on DE, IE, SE (left to right) for increasing sensitivity parameter (top to bottom).

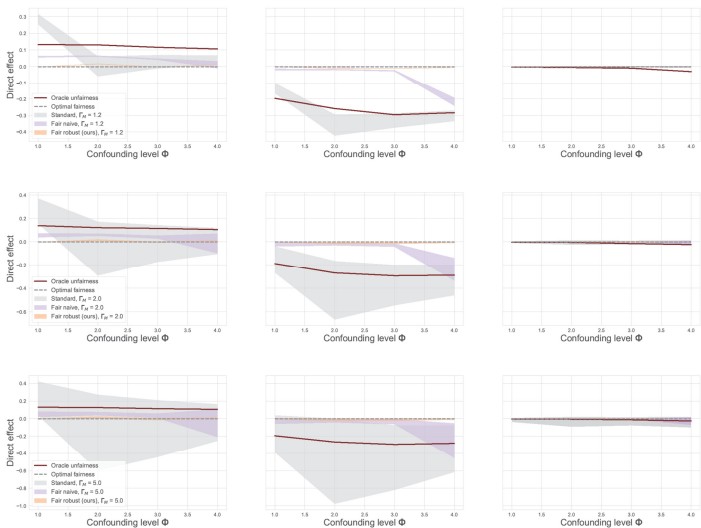

Figure 11: Bounds of the three prediction models on the dataset including *indirect unobserved confounding* on DE, IE, SE (left to right) for increasing sensitivity parameter (top to bottom).

## I.2 PERFORMANCE COMPARISON FOR VARYING SENSITIVITY PARAMETER

We now assess the robustness of our framework with respect to the sensitivity parameter used for training the fair prediction model, i.e., the level of unobserved confounding assumed to be present in the data. To do so, we trained multiple fair regression models on the real-world dataset for increasing sensitivity parameters $\Gamma \in [1, 4]$. In Figure 12, we show the relative increase in performance loss ($\Delta$ MSE) compared to the naïvely fair prediciton model, i.e., $\Gamma = 1$. We observe only a very small increase in performance loss, which demonstrates the robustness of our framework.

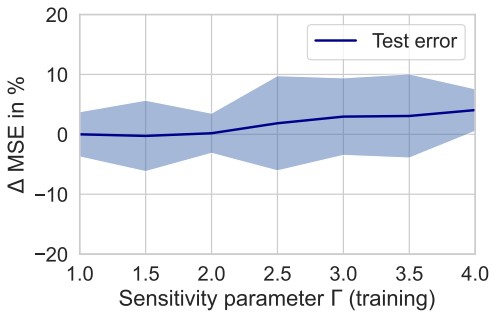

Figure 12: Increase in prediction loss (MSE) in percent for increasing sensitivity parameter chosen for training the fair regressor. Mean and standard deviation calculated over 5 runs.

## J  EXTENSION TO CONTINUOUS FEATURES

For simplicity, we have introduced our framework framework for discrete features in the main paper. Nevertheless, our framework is *directly applicable to continuous features*. The derivations of the bounds are straightforward; they replace the sum over all realizations of $Z$ with the integral. For the direct effect, it follows that

$$
\begin{aligned}
\mathrm{DE}_{a_i,a_j}^{\pm}(y \mid a_i) = {}& \frac{1}{P(a_i)} \int_{\mathcal{Z}} \sum_{m \in \mathbf{M}} P^{\pm}(y \mid m, z, a_j) P^{\pm}(m \mid z, a_i) p(z) dz \\
& - \frac{P(a_j)}{P(a_i)} \int_{\mathcal{Z}} \sum_{m \in \mathbf{M}} P(y \mid m, z, a_j) P(m \mid z, a_i) p(z) dz - P(y \mid a_i),
\end{aligned}
\tag{73}
$$

where $p(z)$ denotes the density of $Z$. The other effects follow in the same manner.

To show the applicability of our framework for continuous features, we perform binary classification on a dataset with multiple continuous features. The dataset is generated as

$Z_1 \sim \mathrm{Uniform}[0.5 - 0.02 U_{\mathrm{SE}}, 1.5 - 0.02 U_{\mathrm{SE}}]$

$Z_2 \sim \mathrm{Uniform}[1 - 0.02 U_{\mathrm{SE}}, 2 - 0.02 U_{\mathrm{SE}}]$

$Z_3 \sim \mathrm{Uniform}[1.5 - 0.02 U_{\mathrm{SE}}, 2.5 - 0.02 U_{\mathrm{SE}}]$

$Z_4 \sim \mathrm{Uniform}[2 - 0.02 U_{\mathrm{SE}}, 3 - 0.02 U_{\mathrm{SE}}]$

$A \sim \mathrm{Bernoulli}(\sigma(0.1 U_{\mathrm{IE}} + 0.1 U_{\mathrm{DE}} + 0.05 U_{\mathrm{SE}} + 0.25 Z_1 + 0.25 Z_2 + 0.25 Z_3 - 0.5 Z_4))$

$M \sim \mathrm{Bernoulli}(\sigma(0.1 Z_1 + 0.1 Z_2 + 0.1 Z_3 - 0.5 Z_4 + 2A - 0.1 U_{\mathrm{IE}})$

$Y = \mathbb{1}_{[(0.1 Z_1 + 0.1 Z_2 + 0.1 Z_3 + 0.1 Z_4 + M + 2A - 0.1 U_{\mathrm{DE}}) \geq 2]}$,

for $U_{\mathrm{DE}}, U_{\mathrm{IE}}, U_{\mathrm{SE}} \sim \mathcal{N}(\Phi, e^{-4})$ with $\Phi \in [1, 4]$.

We train the three classifiers (analogous to our main paper): (1) standard, (2) fair naïve, and (3) fair robust (=our). We train them on the above datasets with sensitivity parameter $\Phi = 2$ and fairness constraints of 0.5 on all effects. For simplicity, we optimize the constraint classifiers with a fixed penalty constraint with weight $\lambda = 2$ instead of augmented lagrangian optimization.

In Figure 13, we present bounds on the three classifiers for sensitivity parameter 2.0. We observe that training the classifier via our framework successfully restricts the bounds to be lower than the constraint level of 0.5. In sum, the results confirm the effectiveness of our framework also in settings with continuous features.

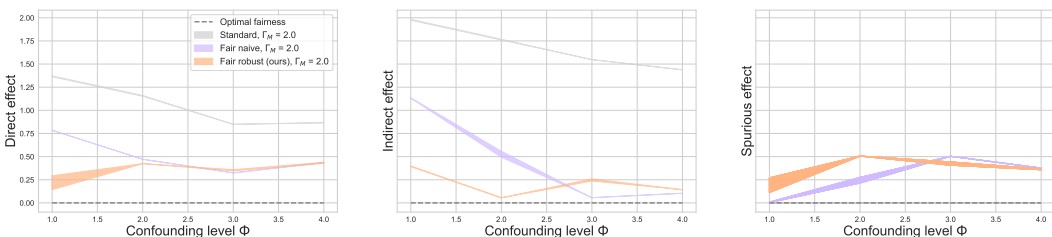

Figure 13: Fairness bounds on the three classifiers on experiments with multiple continuous confounders.

We report the performance of the prediction models in Table 6. We observe that our fair robust model only suffers from a low loss in prediction performance while achieving fairness below the threshold of 0.5 for all three effects.

Beyond continuous features, it may also be interesting to use our framework for continuous mediators. Deriving sharp bounds for continuous mediators in sensitivity analysis is highly non-trivial. Therefore, *no* non-parametric method for such a problem has been developed so far. Nevertheless, this posits an interesting direction for future research. As such, we suggest the following solution approaches:

- Discretization: Discretization is commonly used in causal inference and policy learning to handle continuous variables (e.g., actions in reinforcement learning are commonly discrete to yield continuous values). Discretization may induce an overall performance loss,

Table 6: Performance of the prediction models on the continuous datasets measured by the ROC AUC and the fairness utility.

| Experiment | ROC AUC | | | Fairness utility | | |
|---|---|---|---|---|---|---|
| | Standard | Fair naïve | Fair robust | Standard | Fair naïve | Fair robust |
| $\Phi_{U_{\mathrm{DE}}}, \Phi_{U_{\mathrm{IE}}}, \Phi_{U_{\mathrm{SE}}} = 1$ | 1.0000 | 0.9921 | 0.7048 | $-0.5360$ | $-0.0094$ | $-0.0302$ |
| $\Phi_{U_{\mathrm{DE}}}, \Phi_{U_{\mathrm{IE}}}, \Phi_{U_{\mathrm{SE}}} = 2$ | 1.0000 | 0.9326 | 0.9774 | $-0.5361$ | $-0.0369$ | 0.0737 |
| $\Phi_{U_{\mathrm{DE}}}, \Phi_{U_{\mathrm{IE}}}, \Phi_{U_{\mathrm{SE}}} = 3$ | 1.0000 | 0.9383 | 0.9647 | $-0.5356$ | $-0.0339$ | 0.1158 |
| $\Phi_{U_{\mathrm{DE}}}, \Phi_{U_{\mathrm{IE}}}, \Phi_{U_{\mathrm{SE}}} = 4$ | 1.0000 | 0.8783 | 0.9536 | $-0.5356$ | $-0.0723$ | 0.1072 |

yet, recently, there have been several methods for handling and minimizing this loss (e.g., Rajbahadur et al., 2021). For certain machine learning models, discretization can even increase the model performance (e.g., Lustgarten et al., 2008; Varghese & Sundari, 2022).

- Proxy variables: Another idea is to employ proxy variables instead of sensitive attributes to enforce fairness (e.g., Gupta et al., 2018; Kilbertus et al., 2017) or learning causal effects based on proxies to circumvent unidentifiability due to unobserved confounders (e.g., Miao et al., 2018; Xu & Kanagawa, 2021). Hence, one solution is to incorporate continuous mediators into our framework through a discrete proxy variable of the mediator.

