# OpenReview forum: "Causal Fairness under Unobserved Confounding: A Neural Sensitivity Framework"
_ICLR.cc/2024/Conference — ICLR 2024 poster_

### Official Review · Reviewer_uvD6 · 2023-10-28

**Soundness:** 3 good
**Presentation:** 2 fair
**Contribution:** 2 fair
**Rating:** 5
**Confidence:** 3

**Summary:**

This paper delves into the sensitivity analysis of causal fairness criteria, specifically focusing on counterfactual direct effect (Ctf-DE), indirect effect (Ctf-IE), and spurious effects concerning unobserved confounding. The authors establish bounds for these measures by utilizing the generalized marginal sensitivity model (GMSM) and present a model for learning fair predictions. Experimental results underscore the method's effectiveness to some extent.

**Strengths:**

- The sensitivity analysis on unobserved confounders on causal fairness criterion is a relevant and important research problem.

- The experiments on synthetic data and real data demonstrate the effectiveness of the proposed method at some extent.

**Weaknesses:**

- Limited Contribution and Novelty. The contribution and novelty of this paper may be limited. The theorem presented in the paper, Theorem 1, appears to be a specific application of the Generalized Marginal Sensitivity Model (GMSM) introduced by Frauen et al. (2023), which offers a comprehensive framework for causal sensitivity analysis under unobserved confounding in various settings. It would be beneficial to clarify the distinct contributions and challenges of this work compared to Frauen et al. (2023).

- Limited Scope in Fairness Notions. The paper is somewhat misleading in its claim to perform sensitivity analysis on causal fairness under unobserved confounding. In fact, the focus of this paper is confined to specific causality-based fairness notions based on counterfactual direct effect (Ctf-DE), indirect effect (Ctf-IE) and spurious effects. However, causal fairness encompasses a broader range of notions, such as ones based on proxy discrimination, path-specific causal effects, path-specific counterfactual effects (including natural direct or indirect causal effect), etc. Additionally, it is misunderstanding to call the definitions Ctf-DE, Ctf-IE and Ctf-SE in Zhang & Bareinboim (2018a) as ‘path-specific causal effects’, which differs from their formal definition [1], thus leading to confusion.

- Incomplete Literature Review. The paper lacks a comprehensive review of prior literature on sensitivity analysis on causal effects to unobserved confounding, such as marginal sensitivity model. Additionally, it would be beneficial to provide a brief overview of sensitivity analysis models, including the GMSM, and discuss their strengths and weaknesses.

- Enhanced Experimental Analysis. In the experimental section, it would be advantageous to report and compare prediction performance across various models under different levels of confounding. This would provide a more robust assessment of the proposed method's performance.

- Handling Continuous Variables. The paper assumes that the variables Z and M are discrete, yet many real-world variables are continuous. Therefore, it is essential to discuss how the proposed method can be extended to accommodate continuous variables.

[1] Avin, Chen, Ilya Shpitser, and Judea Pearl. "Identifiability of path-specific effects." (2005).

**Questions:**

The authors state that "A key benefit of the GMSM is that it can deal with discrete mediators and both discrete and continuous outcomes." It is not convincing. It would be helpful to elaborate on other reasons for adopting the GMSM and discuss the strengths and weaknesses of alternative sensitivity models.

---

> ### Author Response · Authors · 2023-11-19
> **Response to reviewer uvD6**
>
> We are grateful for the review! We appreciate the reviewer’s constructive feedback and approved our paper accordingly.
>
> # Response to weaknesses
>
> **1)  	Contribution and novelty:**
>
> Thank you for giving us the opportunity to emphasize the novelty of our framework. Thereby, we clarify that we do **not** simply combine two concepts but that our derivations are **non-trivial**.
>
> *Why is the derivation of our bounds non-trivial?* We have derived tailored bounds for counterfactual fairness under unobserved confounding. Importantly, our bounds require a novel and careful derivation. In other words, our bounds are **not** just a simple result of adopting a sensitivity model but require a new derivation tailored to our setting. The reason is the following: Causal fairness notions commonly contain nested counterfactuals and are thus located on **level three** of Pearl’s ladder of causality. The GMSM (and other sensitivity models) incorporate single-intervention counterfactuals and are thus located on **level two**. Because of that, existing bounds from sensitivity models are **not** applicable. In other words, we can **not** simply adopt existing bounds from sensitivity models, but we need a **new derivation** for level three of Pearl’s causality ladder.
>
> *What is our contribution?* We agree that causal fairness and partial identification bounds under the MSM/GMSM have **separately** been studied before. One of our contributions is the **transfer of concepts** from sensitivity analysis to the causal fairness literature (note: due to the reasons above, this is non-trivial): As a result, we offer a new framework to assess causal fairness under unobserved confounding. This is a major difference to the existing literature on causal fairness, which has ignored the fairness implications due to unobserved confounding. In other words, we fill an important research gap in the literature (see our new Table 1). Furthermore, we provide important implications for practice: we call for a more cautious use of causal fairness since, due to unobserved confounding, important fairness issues may still be present, because of which fairness in practice can be undermined. As a remedy, we contribute for the first time a framework that can remove potential issues in causal fairness due to unobserved confounding. We thus expect that our findings, as well as our framework, are of immediate practical importance.
>
> *How is our framework novel compared to sensitivity analysis?* While we adopt a sensitivity analysis as the basis for deriving our bound, we add over sensitivity analysis in several ways. (1) We derive new bounds that are tailored to our bounds. The bounds do not directly follow from the sensitivity analysis (which is unlike other applications of sensitivity models in the literature); instead, a tailored derivation is needed that is non-trivial. (2) We develop an end-to-end framework for fair prediction models. As such, Steps 1 and 2 in our framework are novel and make large contributions to existing sensitivity analysis.
>
> **Actions:** We improved our work in the following ways to clarify our novelty and why our contributions are **non-trivial**:
>
> * We included a new **Table 1**, where we highlight the novelty of our work over existing literature streams. Importantly, ours is the first work to study causal fairness under unobserved confounding.
> * We revised the presentation of our framework to explain why existing bounds from sensitivity models are **not** applicable (as our causal query is in ladder three and not in ladder two of Pearl’s causality ladder). As a result, we state clearly why our bounds are non-trivial (see revised **Section 2**).
> * We added a technical background explaining the theoretical differences between existing bounds from sensitivity models (=ladder two) and our causal query (=ladder three). Thereby, we motivate why a new derivation is needed (see new **Supplement C.3**).
>
>
>
>
> **2)  	Broader scope across fairness notions:**
>
> Thank you for this important suggestion. Our framework is specially designed to incorporate path-specific effects. Nevertheless, we want to emphasize that our framework is general and can also be applied to other fairness notions. To demonstrate that, we added a new **Supplement F**, where we **derive bounds for other notions**. Thereby, we demonstrate how our framework can be applied to further fairness notions, which shows the broad applicability of our framework.
> Thank you as well for pointing out the ambiguity of the name  “path-specific causal effects”. We rephrased the notation in our paper and now refer to the effects from Zhang & Bareinboim (2018) as “path-specific causal fairness”.
>
> **Action:** We have clarified throughout the paper that our framework is general and can also be applied to other fairness notations. We also added new theoretical results for other fairness notions (see our new **Supplement F**).

---

> ### Author Response · Authors · 2023-11-19
> **Response to reviewer uvD6 (continued)**
>
> **3)  	Extended literature review:**
>
> Thank you. We **extended our literature overview** and now provide a  comprehensive review of prior literature on sensitivity analysis on causal effects to unobserved confounding, such as the marginal sensitivity model. In particular, we added the following materials:
> * We added a comprehensive literature overview on sensitivity analysis (see our new **Supplement C.2**). Therein, we review existing works but also discuss that these are mainly focused on causal effects and not causal fairness. To the best of our knowledge, there is **no** work other than ours that has combined sensitivity analysis and causal fairness. As we lay out, our paper is the **first** to make predictions wrt. causal fairness under unobserved confounding.
> * We added a **discussion of the strengths and weaknesses** of various sensitivity models. Thereby, we highlight the reasons for adopting the GMSM in our work (see our new **Supplement C.2**).
>
>
>
> **4)  Enhanced experimental analysis:**
>
> Thank you. We are more than happy to expand our experimental analysis. We thus included several **new experimental analyses** as follows:
> * We report the prediction performance in **Supplement G**. (Due to space constraints, we do so in the supplements and not in the main paper.)
> * We further show the robustness of our framework. For this, we added additional experimental results on the performance of our fair classifier when trained with different sensitivity parameters (see our new **Supplement I**).
>
>
>
> **5)  	Continuous variables:**
>
> Thank you. We improved our paper to show that our framework is also effective for continuous variables:
> * We clarified that the features (confounders) are **not restricted** to discrete variables.  Instead, we now explain that our framework is **applicable to multiple features that can be binary, continuous, and/or discrete**. Therefore, our method is also applicable to multiple continuous and, thus as well to high-dimensional features (see our revised **Section 3**).
> * We provide **new experimental results** to demonstrate the effectiveness of our framework for multiple continuous variables (see new **Supplement J**).
> * We also discussed an extension of our framework for continuous mediators (see new **Supplement J**).
>
>
>
> # Response to question
>
> **Response to “Benefits of the GMSM”**
>
> Thank you for bringing up this important question. We added a **discussion of the strengths and weaknesses** of various sensitivity models. Thereby, we highlight the reasons for adopting the GMSM in our work (see our new **Supplement C.2**).

---

> ### Comment · Reviewer_uvD6 · 2023-11-22
> **Thanks for the response.**
>
> I thank the authors for responding to all the comments, which addresses some of my concern.
>
> I acknowledge that effectively implementing the sensitivity model necessitates the simplification of causal fairness notions, as defined in Pearl's framework, from level 3 to level 2. This process indeed requires amount of work and careful consideration.
>
> However, as Reviewer rYvN propose, the equation and algorithms is still unclear. This work can be improved from many aspects. To be more clear, the paper structure can be further improved. I hope my comments are helpful for the revision.

---

> > ### Author Response · Authors · 2023-11-22
> > **Follow up on response of reviewer uvD6**
> >
> > Thank you for your response. We are happy that the revision of our paper is well received and that you find our derivation non-trivial.
> >
> > We understand the confusion arosen from our notation, which reviewer rYvN kindly noted, and apologize for the ambiguity of our notation and the impreciseness in Algorithm 1 in the original draft.
> >
> > Therefore, we **addressed all questions** and suggestions for improvements stated by reviewer rYvN regarding the mathematical presentation of our framework. Especially:
> > 1. We **updated Eq. 11**, as well as the introduction in Section 5 to specifically state the non-randomness of the fairness effects and the independence of $y$.
> > 2. We included these **changes in Algorithm 1** to remove the ambiguity of our notation in lines 6-8 (original algorithm).
> > 3. We provided a **new alternative algorithm** (see our **new Algorithm 2** in **new Supplement E.4**) for mutli-class classification, which constrains the three effects on every outcome class separately.
> > 4. We **discussed our training framework in detail** in the new **Supplement E.4**, especially elaborating on the presentation and the calculation of the constraints for each task, i.e. binary classification, multi-class classification, regression, separately.
> >
> > We have uploaded a new version of our **updated manuscript** in which we mark the changes performed based on all of reviewer rYvN’s comments in **red** color.
> >
> > If there are any additional aspects of our presentation that should be improved, we are more than happy to include further suggestions in our manuscript.

---

> ### Comment · Reviewer_uvD6 · 2023-11-23
> **Thanks.**
>
> Thanks for the authors' careful responding to all the comments. I will raise my score.
>
> However, I still think it is not suitable to claim the contributions on causal fairness in the main body (and title), since the paper actually is confined to specific causality-based fairness notions based on counterfactual direct effect (Ctf-DE), indirect effect (Ctf-IE) and spurious effects, instead of the general causal fairness notions. Although the authors added a new Supplement F, deriving bounds for fair on average causal effect and a simple path-specific causal fairness, it does not encompass all causal fairness notions, like counterfactual fairness or path-specific counterfactual fairness. It seems the authors mix the use of some causal fairness notions sometime. For example, it is also confusing to change 'path-specific causal effect' to 'path-specific causal fairness', since actually path-specific causal fairness is defined on path-specific causal effect and they are the same thing in the essence. To avoid confusion and overclaim, I think it is better to reorganize the structure of the paper.
>
> In conclusion, I acknowledge the importance of the performing sensitivity analysis in the causal fairness area, however I still do not think this paper is ready to be published and once again remark that I hope to see an improved version of your work published in the future.

---

> > ### Author Response · Authors · 2023-11-23
> > **Follow up on response of reviewer uvD6**
> >
> > Thank you for the constructive discussion and for raising your score! Please allow us to elaborate on your remaining concern, which we believe is addressed by a simple rewriting of our contributions.
> >
> > - *Causal fairness notions:* Thank you for pointing out the ambiguity in our wording. Please note that, by naming our fairness notions “causal fairness,” we are consistent with previously established literature [1]. However, we agree that the naming “causal fairness” may lead to confusion as it suggests that other fairness notions based on causal theory (e.g., counterfactual fairness) are included.
> >
> > **Action:** We have added a clarification on the naming of different fairness notions and spelled out we do not consider other fairness notions based on causal theory (see the red text in our revised introduction).
> >
> > - *Why do we not consider other fairness notions based on causality?* We agree that,  generally, other fairness notions can also be of interest. The reason why we specifically focus on “causal fairness” [1] is that these fairness notions are not only practically relevant but also **allow for counterfactual unnesting without strong assumptions** (which is what we prove in Theorem 1). That is, we can first reduce the corresponding layer 3 query to a layer 2 query and subsequently leverage results from (layer 2-based) sensitivity analysis. For other fairness notions, such as counterfactual fairness, **such unnesting is impossible without strong assumptions and fundamentally different approaches**. For example, it is well known that **counterfactual fairness is not even identifiable in settings without unobserved confounding** [1]. Existing approaches require both knowledge of the underlying causal graph and the parametric form of the causal mechanisms (e.g., additive noise models, [2]).  Hence, we argue that we do a first but important step in the direction of using some causal fairness notions in settings with unobserved confounding. Furthermore, **we are consistent with most of the literature that combines causality and fairness in that we only focus on a single type of fairness** (e,g, [2], [3], [4]), for which we provide both novel and rigorous results in the form of sensitivity-based bounds and a practical learning algorithm.
> >
> >
> > [1]	Drago Plecko and Elias Bareinboim. Causal fairness analysis: A causal
> >   	toolkit for fair machine learning. arXiv preprint, 2022.
> >
> > [2]	Niki Kilbertus, Philip J. Ball, Matt J. Kusner, Adrian Weller, and Ricardo
> > 	Silva. The sensitivity of counterfactual fairness to unmeasured confounding. In
> >            	Conference on Uncertainty in Artificial Intelligence, 2019.
> >
> > [3]	Yoichi Chikahara, Shinsaku Sakaue, Akinori Fujino, and Hisashi Kashima. Learning
> > 	individually fair classifier with path-specific causal-effect constraint. In International
> > 	Conference on Artificial Intelligence and Statistics (AISTATS), 2021
> >
> > [4]	Junzhe Zhang and Elias Bareinboim. Fairness in decision-making – the causal
> > 	explanation formula. In Conference on Artificial Intelligence (AAAI), 2018.

---

> > > ### Comment · Reviewer_uvD6 · 2023-11-23
> > >
> > > Thanks. I still think that the paper could benefit from more focused organization to enhance its clarity and coherence.
> > >
> > > Additionally, regarding the page limit issue, it's a concern. I downloaded the original draft, which even violates the page limit. Adhering to submission guidelines, including page limits, is crucial for maintaining fairness in the review process. I do not think this paper could be accepted in this situation.

---

### Official Review · Reviewer_2FkR · 2023-10-31

**Soundness:** 3 good
**Presentation:** 2 fair
**Contribution:** 3 good
**Rating:** 6
**Confidence:** 3

**Summary:**

The paper establishes the partial identification bound for causal fairness under GMSM, which can be used for learning causally fair predictors under specified GMSM and SCM. Authors provide theoretical guarantee for the partial identification bound and demonstrate its benefit using synthetic and real-world data.

**Strengths:**

I find the paper interesting and easy to read. It establishes the partial identification bound for causal fairness under GMSM, which can be used for learning causally fair predictors under specified GMSM and SCM.

I believe the problem studied is important and has important practical value and the paper is theoretically sound and made a step toward better estimation of causal fairness.

**Weaknesses:**

The motivation and solution seem disconnected. Authors suggest practitioners can audit the unobserved confounding in the data in the introduction, but the solution using GMSM requires prior knowledge of UC strength. In this case, we need to know the unobserved confounding strength in order to estimate or optimize the causal fairness.

The novelty is relatively low. The causal fairness and the partial identification bound under MSM/GMSM is extensively studied, the paper simply combined them.

Authors did not report accuracy measures in the paper, which should also be reported. I assume such fairness constraint would impact the prediction accuracy a lot.

A natural criticism is the method is restricted to discrete features, while in practice, variables like crime history may be highly contextual (videos, images, text, ..) . Can authors discuss the difficulty in extending it to high dimensional features?

Can authors explain more on "Fairness bounds in non-identifiable settings": why general non-identifiability in these papers does not encompass the unobserved confounding case? Nabi & Shpitser, 2018 also uses unobserved confounding as one of the examples.

Typo: abstract, sources of unobserved confounding? "ours is the first work to study causal fairness under observed confounding." miss-specification.

Writing: Why repeat the contribution at the end of section 1. The research gap seems to repeat contribution again and such gap is already mentioned before.

**Questions:**

See weakness.

---

> ### Author Response · Authors · 2023-11-19
> **Response to reviewer 2FkR**
>
> Thank you for your positive evaluation of our paper! We took all your comments at heart and improved our paper accordingly.
>
> **Response to (1):** Connection of motivation and solution
>
> We thank the reviewer for the question and are happy to clarify the connection between our motivation and the proposed solution. In practice, knowledge of the unobserved confounding (UC) strength is beneficial but not completely necessary. Importantly, our framework is of direct help in practice when (1.) the maximum UC strength is known or (2.) unknown. We discuss both use cases below:
>
> 1. There are often good reasons why the UC strength is known or why it can be upper-bounded. Many fairness-relevant applications are rooted in social science, where there is a good understanding of potential causes for biases (e.g., around gender, age, etc.). Hence, it is often reasonable to make assumptions that the UC strength is in a specific relationship with some other observed variable [1, 2, 3, 4], e.g., not as strong as (a multiple of) the observed variable. Notwithstanding, several approaches have been proposed on how sensitivity parameters can be chosen in practice. Here, we refer to the discussions in [5, 6, 7, 8].
> 2. Even when the UC strength is unknown, our framework is of significant value in practice. In this case, practitioners can employ our bounds to audit the unobserved confounding in the data. To do so, one can use our framework to calculate bounds for increasing sensitivity parameters until the interval defined by the bounds contains a certain value of interest, e.g., zero (indicating complete fairness). The minimal sensitivity parameter to achieve this goal corresponds to the level of unobserved confounding, which would be necessary to consider the data fair. We can also view the sensitivity parameter as our uncertainty about the fairness of our prediction model. Smaller sensitivity parameters not achieving the goal still provide information about the sign, i.e., direction, of the unfair effect [6]. Generally, testing various sensitivity parameters is a common technique in real-world applications of sensitivity analysis [5, 9, 10]. In sum, even when the UC strength is unknown, our framework can thus be of large practical value.
>
> **Actions:** We have carefully revised our paper in the following ways:
> * We have improved our motivation to clarify the connection to our solution framework (see revised introduction).
> * We added a discussion on the practical considerations from above (see new **Supplement B.2**).

---

> > ### Author Response · Authors · 2023-11-19
> > **Response to reviewer 2FkR (continued)**
> >
> > **Response to (2):** Novelty and why bounds are non-trivial
> >
> > Thank you for giving us the opportunity to emphasize the novelty of our framework. Thereby, we clarify that we do **not** simply combine two concepts but that our derivations are **non-trivial**.
> >
> > *Why is the derivation of our bounds non-trivial?* We have derived tailored bounds for counterfactual fairness under unobserved confounding. Importantly, our bounds require a novel and careful derivation. In other words, our bounds are **not** just a simple result of adopting a sensitivity model but require a new derivation tailored to our setting. The reason is the following: Causal fairness notions commonly contain nested counterfactuals and are thus located on **level three** of Pearl’s ladder of causality. The GMSM (and other sensitivity models) incorporate single-intervention counterfactuals and are thus located on **level two**. Because of that, existing bounds from sensitivity models are **not** applicable. In other words, we can **not** simply adopt existing bounds from sensitivity models, but we need a **new derivation** for level three of Pearl’s causality ladder.
> >
> > *What is our contribution?* We agree that causal fairness and partial identification bounds under the MSM/GMSM have **separately** been studied before. One of our contributions is the **transfer of concepts** from sensitivity analysis to the causal fairness literature (note: due to the reasons above, this is non-trivial): As a result, we offer a new framework to assess causal fairness under unobserved confounding. This is a major difference from the existing literature on causal fairness, which has ignored the fairness implications due to unobserved confounding. In other words, we fill an important research gap in the literature (see our new Table 1). Furthermore, we provide important implications for practice: we call for a more cautious use of causal fairness since, due to unobserved confounding, important fairness issues may still be present, because of which fairness in practice can be undermined. As a remedy, we contribute for the first time a framework that can remove potential issues in causal fairness due to unobserved confounding. We thus expect that our findings, as well as our framework, are of immediate practical importance.
> >
> > *How is our framework novel compared to sensitivity analysis?* While we adopt a sensitivity analysis as the basis for deriving our bound, we add over sensitivity analysis in several ways. (1) We derive new bounds that are tailored to our bounds. The bounds do not directly follow from the sensitivity analysis (which is unlike other applications of sensitivity models in the literature); instead, a tailored derivation is needed that is non-trivial. (2) We develop an end-to-end framework for fair prediction models. As such, Steps 1 and 2 in our framework are novel and make large contributions to existing sensitivity analysis.
> >
> > **Actions:** We improved our work in the following ways to clarify our novelty and why our contributions are **non-trivial**:
> > 1. We included a new **Table 1**, where we highlight the novelty of our work over existing literature streams. Importantly, ours is the first work to study causal fairness under unobserved confounding.
> > 2. We revised the presentation of our framework to explain why existing bounds from sensitivity models are **not** applicable (as our causal query is in ladder three and not in ladder two of Pearl’s causality ladder). As a result, we state clearly why our bounds are non-trivial (see revised **Section 2**).
> > 3. We added a technical background explaining the theoretical differences between existing bounds from sensitivity models (=ladder two) and our causal query (=ladder three). Thereby, we motivate why a new derivation is needed (see new **Supplement C.3**).
> >
> >
> >
> > **Response to (3):** Reporting of performance measures
> >
> > Thank you. We are more than happy to report the prediction performance in **Supplement G**.
> >
> >
> >
> >
> > **Response to (4):** Extension to high-dimensional features
> >
> > Thank you. We improved our paper to show that our framework is also effective for high-dimensional features:
> > * We clarified that the features (confounders) are **not restricted** to discrete variables. Instead, we now explain that our framework is **applicable to multiple features that can be binary, continuous, and/or discrete**. Therefore, our method is also applicable to multiple continuous and, thus as well as to high-dimensional features (see our revised **Section 3**).
> > * We provide **new experimental results** to demonstrate the effectiveness of our framework for multiple continuous variables (see new **Supplement J**).
> > * We also discussed an extension of our framework for continuous mediators (see new **Supplement J**).

---

> > > ### Author Response · Authors · 2023-11-19
> > > **Response to reviewer 2FkR (continued)**
> > >
> > > **Response to (5):** Difference to general non-identifiability
> > >
> > > Thank you. We appreciate the opportunity to clarify the **difference** between our framework for unobserved confounding and existing work on general non-identifiability. Of note, both are **different** concepts. Unobserved confounding is only one reason for non-identifiability. Other reasons are, e.g., interference or lack of overlap. Methods able to provide fair predictions in general non-identifiable settings are thus not tailored to unobserved confounding specifically, resulting, e.g., in inefficient fairness constraints.
> > >
> > >
> > > The paper by Nabi & Shpitser [11] provides an excellent motivation for our paper. Therein, the authors state that causal effects are unidentifiable in the presence of unobserved confounding, and, as a result, causal fairness may be violated due to unobserved confounding. However, the paper by Nabi & Shpitser [11] provides merely a theoretical conceptualization of the problem but **without an actual algorithm** to solve the problem. The latter (i.e., an algorithm to mitigate issues from unobserved confounding for casual fairness) is our novelty. Because of that, Nabi & Shpitser [11] is also **not** applicable as a baseline. In sum, the paper by Nabi & Shpitser is **vastly different from ours**: their paper does **not** quantify the impact on fairness from unobserved confounding, and it does mitigate fairness issues due to unobserved confounding. In contrast, only our paper **provides fairness bounds** and **a framework** for training fair prediction models in the presence of unobserved confounding.
> > >
> > > **Response to (6) & (7):** Typo & Writing
> > >
> > > Thank you. We fixed the text accordingly (see blue text in Section 1).
> > >
> > >
> > >
> > > **References:**
> > >
> > > [1]	Carlos Cinelli, and Chad Hazlett. "Making sense of sensitivity: Extending omitted variable bias." Journal of the Royal Statistical Society Series B: Statistical Methodology 82.1, 2020, pp. 39-67.
> > >
> > > [2]	Alexander M. Franks, Alexander D’Amour, and Avi Feller. "Flexible sensitivity analysis for observational studies without observable implications." Journal of the American Statistical Association 115.532, 2019, pp. 1730-1746.
> > >
> > > [3]	Matthew A.Masten, and Alexandre Poirier. "Inference on breakdown frontiers." Quantitative Economics 11.1, 2020, pp. 41-111.
> > >
> > > [4]	Zhongheng Zhang, et al. "Causal mediation analysis in the context of clinical research." Annals of translational medicine 4.21, 2016.
> > >
> > > [5]     	Ying Jin, Zhimei Ren, and Emmanuel J. Candès. “Sensitivity analysis of individual treatment effects: A robust conformal inference approach”. In Proceedings of the National Academy of Sciences (PNAS) 120.6, 2023.
> > >
> > > [6]     	Nathan Kallus, Xiaojie Mao, and Angela Zhou. “Interval estimation of individual-level causal effects under unobserved confounding”. In Conference on Artificial Intelligence and Statistics (AISTATS), 2019.
> > >
> > > [7] 	Iván Díaz, and Mark J. van der Laan. "Sensitivity analysis for causal inference under unmeasured confounding and measurement error problems." The international journal of biostatistics 9.2, 2013, pp. 149-160.
> > >
> > > [8]	Kosuke Imai, and Teppei Yamamoto. "Identification and sensitivity analysis for multiple causal mechanisms: Revisiting evidence from framing experiments." Political Analysis 21.2, 2013, pp. 141-171.
> > >
> > > [9]     	Jesse Y. Hsu and Dylan S. Small. “Calibrating sensitivity analyses to observed covariates in observational studies”. In Biometrics 69.4, 2013, pp. 803–811.
> > >
> > > [10]	Tyler J. VanderWeele, and Ding Peng. "Sensitivity analysis in observational research: introducing the E-value." Annals of internal medicine 167.4, 2017, pp. 268-274.
> > >
> > > [11]     	Razieh Nabi and Ilya Shpitser. “Fair inference on outcomes”. In Conference on Artificial Intelligence (AAAI), 2018.

---

> > > > ### Comment · Reviewer_2FkR · 2023-11-23
> > > > **Response to rebuttal**
> > > >
> > > > Thank authors for the detailed feedback. I think authors address my concern on the differences about previous work and contribution.
> > > >
> > > > On the motivation, I think we have different understanding of the term "auditing". When the authors mentioning auditing unobserved confounding, my initial understanding is the method can infer when unobserved confounding happens. Based on the rebuttal, I can see that the authors means auditing the minimum level of unobserved confounding needed to achieve a specified counterfactual fairness level. This form of auditing is unusual (to me) and I still don't understand why it is useful in practice, especially when we do not know the actual unobserved confounding level (when we know, there is also no need for this kind of auditing), I hope authors can be more clear in the final version about its definition and merit practically.
> > > >
> > > > Another weakness from the additional experimental results is the fair robust algorithm has a very poor predictive performance in terms of prediction accuracy. In the new result in supp G that authors did not report in the first draft, algorithms are close to random guessing in some settings. While it is expected that the accuracy would drop, it is still of concern that it is just as good as random guessing.
> > > >
> > > > Overall, I still think the paper is correct and clear and address an important problem and I will keep my score.

---

> > > > > ### Author Response · Authors · 2023-11-23
> > > > > **Follow up on response of reviewer 2FkR**
> > > > >
> > > > > Thank you for your response. We are happy that our clarifications could address your concern about our contribution.
> > > > >
> > > > > We regret that the wording in our motivation (“auditing”) caused misunderstandings. To prevent future misunderstandings, we have revised our wording, and refer to our framework as a **refutation strategy**, which examines the worst-case (un-)fairness of a model instead of a method for auditing prediction models. Refutation strategies [1] are common in traditional causal inference to validate estimates from causal estimators, especially wrt. the underlying assumptions and our work contributes by presenting such a refutation strategy in the context of causal fairness. We want to highlight again that we **do not audit/examine** the prediction model regarding **unobserved confounding** but regarding the **fairness** of the model.
> > > > >
> > > > > **Action:** We have revised our above wording (in red color) following your suggestions.
> > > > >
> > > > > Furthermore, we would like to **clarify additional aspects of the motivation**:
> > > > > Making inferences about when unobserved confounding occurs (and when not) is **not possible** since the **no unobserved confounding** assumption in causal inference is **not testable**. This is a widely stated fact in the causal inference literature. Nevertheless, in real-world applications, it is highly unlikely to observe all confounders, and therefore, models almost always suffer from unobserved confounding (and will thus have confounding bias!). As a result, causal effects and, thus, causal fairness are not directly identifiable. One way to partially estimate effects in the presence of unobserved confounders is through causal sensitivity analysis, which requires the specification of a sensitivity parameter. Even **if the UC strength is known**, and thus, the proper sensitivity parameter can be specified, we still **need** to employ a **sensitivity model** to identify causal effects and fairness. If the UC strength is unknown, our framework is still of significant value in practice, as outlined in our previous answer. Therefore, knowing the UC strength does not diminish the need for auditing a prediction model with respect to fairness.
> > > > >
> > > > > We agree that the prediction performance of fair models varies depending on the task (and the level of unobserved confounding). If the underlying data is unfair, well-fitting prediction models inherit the unfairness of the data directly. Therefore, there always exists a **trade-off** between prediction **performance and fairness**. We tested our framework in settings with different levels of unobserved confounding, including settings in which even unconstraint models do _not_ achieve high prediction performance. Practitioners can use the prediction performance as a “risk signal” to understand when practical applications are prone to large unobserved confounding and when unobserved confounding may undermine unfairness. If the unconstraint task is easy to learn, the performance of the fair classifier is still satisfactory. We further see that the trade-off is highly problem-dependent. To show that, we have now additionally added the prediction performance for our continuous experiments (see our new materials in **Supplement J**). Therein, we see that the performance is still **satisfactory**.
> > > > >
> > > > > **Action:** We added the prediction performance to our experiments in our continuous setting (see new results in red color in **Supplement J**), where we find that the performance is satisfactory.
> > > > >
> > > > > References
> > > > >
> > > > > [1] Sharma, A., Syrgkanis, V., Zhang, C., & Kıcıman, E. (2021). Dowhy: Addressing challenges in expressing and validating causal assumptions. arXiv preprint arXiv:2108.13518.

---

### Official Review · Reviewer_rYvN · 2023-11-12

**Soundness:** 3 good
**Presentation:** 3 good
**Contribution:** 2 fair
**Rating:** 6
**Confidence:** 4

**Summary:**

The paper integrates a recent framework for sensitivity analysis within a common causal fairness family.

**Strengths:**

The paper is nicely laid out and mostly easy to follow. The problem is important, as sensitivity analysis plays a substantive role in a variety of causal estimands. End-to-end learning to produce fair classifiers is presented, illustrating that sensitivity analysis doesn't need to take place just for standard causal estimands such as average causal effect.

**Weaknesses:**

May main comment is to what extent this mostly is a direct application of Frauen et al. (2023). If I understood it well, is the main technical innovation the use of Lemma 1 in the context of GMSM? Or is this even more closely related to the cited paper?

I’m confused by Eq. (11): there is an expectation over $Y$. Why is this form of aggregation sensible? I’d expect that we would like to enforce the constraint uniformly over all possible values in the same sample space of $Y$ instead of  performing any sort of probabilistic averaging. Speaking of which, which distribution are we marginalizing over, is this the marginal distribution of $Y$? What happens to $a_i$, $a_j$ from (1)-(3)?

Maybe this is well-explored in the causal fairness literature, but the switch between “real Y” and $f_\theta$ in the training procedure in Section 5 is very confusing: we design $f$, so any confounding between “$Y$ “ (that is, $f_\theta$) vanishes by design. Why would we consider any sort of sensitivity parameter for “$Y$”?

I’m worried about the theoretical results. In Appendix D.3 the authors state that they “extensively use the following corollary”, which boils down to Eq. (33). This equation refers to “$P(X |do(A=a), A\neq a)$”. The event past the conditioning bar has probability zero, as the control signal $do(A = a)$ implies $A = a$ (yes, we can condition on measure-zero events, that’s what we do for continuous variables, but even then we take for granted the careful textbook measure-theoretical characterization of this conditional that we implicitly accept by default, but that’s a non-trivial (and not unique!) characterization of conditioning. But  (33) doesn’t appear to make sense even for discrete $A$). I may be missing something obvious, in which case I will appreciate a clarification – if this indeed “follows directly from basic probability theory” I’m happy to be taught the baby steps explicitly…  Of course we can say that

$$P(X | do(A = a)) = P(X(a)) = P(X(a) | A = a)P(A = a) + P(X(a) | A \neq a)P(A \neq a) = $$
$$P(X | A = a)P(A = a) + P(X(a) | A \neq a)P(A \neq a) , $$

but $P(X(a) | A \neq a)$ is not the same thing as $P(X | do(A = a), A \neq a)$. Among other things, $X$ is not $X(a)$ when $A \neq a$, but it is undefined what $X$ even means under regime $do(A = a)$ and event $A \neq a$.

**Questions:**

This is. summary of the above.

- What is the novelty?

- Why averaging over $Y$ in Eq. (11) makes sense (and which distribution are we averaging over)?

- Is there any point in prescribing sensitivity analysis between $Y$ and $A$, given that $Y$ is not involved in the prediction?

- Explaining better the meaning of Eq. (33) and where it is used.

I would also would like to know whether any path-specific effect could be tackled and not only the three default ones.

---

> ### Author Response · Authors · 2023-11-19
> **Response to reviewer rYvN**
>
> Thank you for your very constructive and detailed review of our paper! We took all your comments at heart and improved our paper accordingly.
>
>
> **Response to (1):** Difference from Frauen et al (2023)
>
> Thank you for giving us the opportunity to emphasize the novelty of our framework. Thereby, we clarify that we do **not** have a simple application of Frauen et al. (2023) but that our derivations are **non-trivial**.
>
> *Why is the derivation of our bounds non-trivial?* We have derived tailored bounds for counterfactual fairness under unobserved confounding. Importantly, our bounds require a novel and careful derivation. In other words, our bounds in Theorem 1 are **not** just a simple result of applying Frauen et al. (2023) but require a new derivation tailored to our setting. The reason is the following: Causal fairness notions commonly contain nested counterfactuals and are thus located on **level three** of Pearl’s ladder of causality. The GMSM in Frauen et al. (2023) incorporates single-intervention counterfactuals and is thus located on **level two**. Because of that, existing bounds from sensitivity models are **not** applicable. In other words, we can **not** simply adopt existing bounds from sensitivity models, but we need a **new derivation** for level three of Pearl’s causality ladder.
>
> *What is our contribution?* We use the GMSM from Frauen et al (2023) primarily to formalize our setting, while the actual derivation for our bounds is new. Hence, our derivation does not simply follow from the GMSM but is non-trival (i.e., we only use the GMSM from Lemma 1 to formalize our setting while our Theorem 1 and its derivation is new). Our main contributions are beyond the GMSM. (1) We transfer concepts from sensitivity analysis to causal fairness literature. This is a major difference from the existing literature on causal fairness, which has ignored the fairness implications due to unobserved confounding. As such, we fill an important research gap in the literature (see our new **Table 1**). (2) We propose a new neural framework to assess causal fairness under unobserved confounding and learn robust predictions. Our is the first framework to mitigate fairness violations from unobserved confounding so that still fair predictions are made. (3) We offer new insights for practice: one of our implications is that there can be important fairness uses in practice due to unobserved confounding, because of which fairness in practice can be undermined. We thus expect that our findings, as well as our framework, are of immediate practical importance.
>
> *How is our framework novel compared to sensitivity analysis?* While we adopt a sensitivity analysis as the basis for deriving our bound, we add over sensitivity analysis in several ways. (1) We derive new bounds that are tailored to our bounds. The bounds do not directly follow from the sensitivity analysis (which is unlike other applications of sensitivity models in the literature); instead, a tailored derivation is needed that is non-trivial. (2) We develop an end-to-end framework for fair prediction models. As such, Steps 1 and 2 in our framework are novel and make large contributions to existing sensitivity analysis.
>
> **Actions:** We improved our work in the following ways to clarify our novelty and why our contributions are **non-trivial**:
> 1. We spell out clearly that our setting is grounded in Frauen et al. However, our derivations are new and non-trivial (see revised **Section 2**). We further explain why existing bounds from Frauen et al. are **not** applicable (as our causal query is in ladder three and not in ladder two of Pearl’s causality ladder).
> 2. We added a technical background explaining the theoretical differences between existing bounds from sensitivity models (=ladder two) and our causal query (=ladder three). Thereby, we motivate why a new derivation is needed (see new **Supplement C.3**).
> 3. We included a new **Table 1**, where we highlight that our work is orthogonal to sensitivity analysis and that we make important contributions.

---

> ### Author Response · Authors · 2023-11-19
>
> **Response to (2):** Eq. (11)
>
> Thank you. We have now improved the notation in Eq. (11). Note that the prediction model returns **class probabilities** for classification or the **delta distribution** (point distribution) for regression tasks. Therefore, $P(Y)$ does not correspond to the true distribution of the outcome variable $Y$ but rather to the conditional distribution over the prediction outcomes $P(\hat{Y} \mid z, m, a)$, where $\hat{Y} \mid z, m, a$ defines the prediction outcome, i.e., the predicted distribution of $Y$, for a given realization of  $Z, M$ and $A$. To clarify this, we have updated our notation accordingly and provided further clarifications in the text.
>
> Taking the **expectation** over the class probabilities or the delta distribution is **a natural approach** in binary classification and regression problems. For multiclass classification, other distribution quantiles are separate constraints per output class might also be reasonable depending on the specific requirements of the use case.
>
> **Action:** We have revised Eq. (11) along with the suggestions in your comment.
>
>
>
>
>
> **Response to (3):** Sensitivity analysis between $A$ and $Y$
>
> Thank you for your question. Indeed, it is **not necessary to perform sensitivity analysis** on the relationship between **$A$ and $Y$** since the confounding on the relationship vanishes by design. This means that we do not need to define a sensitivity model for $Y$, i.e., we do not need to specify $\Gamma_Y$. Nevertheless, we still **need a sensitivity model for $M$** to obtain bounds $P^+(m \mid a, z), P^-(m \mid a, z)$ for $P(m \mid a, z)$.
>
> The **sensitivity models** that need to be specified **do not directly map** to the three path-specific causal **fairness effects** (direct, indirect, spurious). They only provide bounds on a given probability. Since all three effects contain $P(m \mid a, z)$, we obtain bounds for all effects, not only for the indirect effect.
>
> If we want to assess the **dataset fairness**, we need to bound $P(y \mid a, z, m)$ and $P(m \mid a, z)$. Therefore, in this case, we need to specify **two sensitivity models**, i.e. one with $\Gamma_M$ and one with $\Gamma_Y$.
>
>
> **Response to (4):** Eq. (33)
>
> Thank you for the suggestion on how to improve our notation (previous: Eq. 33; now Eq. 32). We did not intend to condition on measure-zero events but to **intervene on conditional events**, i.e., set the sensitive attribute to $a_j$ by intervention for events in which the realization was $a_i$. We have thus revised our notation. Reassuringly, we emphasize that all subsequent applications of the equation were correct.
>
> **Action:** We have **rewritten** the equation (now: Eq. 32, previous: Eq. 33) in terms of counterfactual expressions to avoid misleading notation.
>
>
>
>
> **Response to (5):** Applicability to other path-specific effects
>
> Thank you. Our framework is specially designed to incorporate path-specific effects. Nevertheless, we want to emphasize that our framework is general and can also be applied to other fairness notions. To demonstrate that, we added a new **Supplement F**, where we **derive bounds for other notions**. Thereby, we demonstrate how our framework can be applied to further fairness notions, which shows the broad applicability of our framework.
>
> **Action:** We have clarified throughout the paper that our framework is general and can also be applied to other fairness notations. We also added new theoretical results for other fairness notions (see our new **Supplement F**).

---

> ### Comment · Reviewer_rYvN · 2023-11-20
> **Thank you, and a follow up**
>
> Thank you, while I do think that my statement that the main idea is the application of Lemma 1 to the existing bounds of GMSM was correct (Remark 1 is basically the main idea, right?), I acknowledge that by looking at the derivation in the appendix in more detail I agree that it still required quite some work. Even digesting and presenting Correa et al. into this context is a non-trivial effort, which I recognize.
>
> It's frustrating that the original pdf is not available anymore (or maybe I'm just incompetent in trying to find it), but wasn't Eq. 11 just $\mathbb E[Y]$ originally? My main point was that averaging over $A$ and $M$ doesn't seem to make sense to me (imagine that we want to optimize some $\mathbb E[f(Y, A, M)]$ where we need $f(y, a, m)$ to be positive for all $a$, $m$, but we only enforce $\mathbb E[f(Y, A, M)]$ to be positive...). The "max" in the new Eq. 11 is ambiguous. Can you clarify what is this a maximum over what? All $m, a$? (across $CF+$ and $CF-$). I'm not sure whether the iterations in Algorithm 1 are doing that.

---

> > ### Author Response · Authors · 2023-11-20
> > **Response to follow up of reviewer rYvN**
> >
> > Thank you for your fast response. We are happy that the revision of our paper is well received and that you find our derivation non-trivial. We welcome the opportunity to address your follow-up questions.
> >
> > It is unfortunate that the original PDF is no longer visible. We experienced the same behavior in OpenReview. For transparency, we have thus uploaded the original PDF here: *[https://anonymous.4open.science/r/FairSensitivityAnalysis_anonymous-0DA2/README.md]*
> >
> > **Response to your question regarding Eq. (11)**
> >
> > Before we respond to your question, let us first state the old and the new version of Eq. (11) below. The **old version** of Eq. (11) in the original version of our submission was:
> >
> > $ \max\{|\mathbb{E}\_{Y} [\mathrm{CF}^+]|, |\mathbb{E}\_{Y} [\mathrm{CF}^-]|\} \leq \gamma\_{\mathrm{CF}}$
> >
> > We updated it to the **new version**:
> >
> > $ \max\{|\mathbb{E}\_{\hat{Y}\mid z, m, a} [\mathrm{CF}^+]|, |\mathbb{E}\_{\hat{Y}\mid z, m, a} [\mathrm{CF}^-]|\} \leq \gamma\_{\mathrm{CF}}$
> >
> > As you can see, we do **not** average over $A$ and $M$ but over the **conditional** distribution over the prediction outcomes $P(\hat{Y} \mid z, m, a)$ for fixed input samples, i.e. fixed realizations of $M=m, A=a, Z=z$. Note that the CFs are defined for fixed realizations of $A$ and $Y$. For each combination of the realizations, we aim to bound the maximum absolute value of the resulting CFs. We assume the confusion might arise due to the variable $c$ in our pseudo-code. Note that this variable is not a scalar value but a vector of dimension *# constraints*. The same holds for $\lambda$ and $\gamma$.
> >
> > The maximum takes the absolute value of ${\mathbb{E}}\_{\hat{Y}\mid z, m, a} [\mathrm{CF}^+]$ or $\mathbb{E}\_{\hat{Y}\mid z, m, a} [\mathrm{CF}^-]$  for fixed $a, m, z$ (for each $CF \in$  {$DE, IE, SE$ }). Through this definition, we enforce the **minimization of the maximum deviation from zero** (=fair CF), and, therefore, both bounds are driven towards zero in Algorithm 1.
> >
> > **Our main idea is the application of Lemma 1**
> >
> > As you nicely summarized, the main high-level idea of our work is summarized in Remark 1. Nevertheless, we want to emphasize again that our derivations of the bounds and our implementation in a neural framework are _highly non-trivial_.
> >
> > To better navigate between existing literature and our contribution, we have also carefully revised our paper as follows: (1) We have updated the presentation of our framework (see Section 4) to transparently present what is known, what is novel, and where the technical challenges come from. (2) We now refrain from framing the GMSM as “Step 1” of our framework. Instead, we now refer to it as the underlying empirical setting (see revised Section 4.1). (3) We have redrawn the framework in Fig. 3 to make clear that the GMSM is from the literature, while Steps 1 and 2 are our main novelty.

---

> > > ### Comment · Reviewer_rYvN · 2023-11-21
> > > **Follow up**
> > >
> > > Thanks for the follow up, but I'm afraid this is still not clear. For concreteness, let's say:
> > >
> > > * we have some input $X$, output $Y$ all discrete
> > > * the model is a full distribution of $Y$ given $X$ parameterized by some $\theta$
> > > * the loss is negative log-likelihood
> > > * we want the functional $g(p_{y|x}(\cdot | \cdot))$ to be uniformly bounded by some $\gamma$.
> > >
> > > Then what I would write (in population, with the true pmf being $p_0(y | x)$) is
> > >
> > > Minimize (with respect to $\theta$)  $$l(\theta) = -\sum_{y, x}p_0(y, x) \log p_\theta(y | x)$$
> > > subject to $$\forall y, x |g(p_\theta(y | x))| \leq \gamma $$
> > >
> > > (the above is the explicit way of declaring $max |g(p_\theta(y | x))| \leq \gamma $. Don't we need a different Lagrange multiplier for each possible value of $x$ and $y$ in their domain if you are indeed using $L_\infty(|g(p_\theta(\cdot | \cdot) - \gamma|)$ as your constraint?
> > >
> > > I don't see how Algorithm 1 is doing the equivalent of this at all: the max function in line 6 is unclear (a maximum over the sample?), it's not differentiable and it's not clear whether the dependency of $\theta_l$ is carried over to line 8.
> > >
> > > Do the Langrange multipliers you get end up being (close to) zero in the end in the optimization?

---

> ### Author Response · Authors · 2023-11-22
> **Response to follow-up 2**
>
> Thank you for seeing our paper in a good light. We welcome your follow-up questions and are more than happy to answer them below. We acknowledge that our notation was ambiguous ($Y\mid m,a,z$) and hope that our **responses** and **updates in our manuscript** can address your questions and concerns successfully. Specifically, we introduced the notation of expected fairness effects ($\mathrm{CF}\_{\mathbb{E}}$) to clarify the notation.
>
> **General clarifications:**
> All fairness effects are defined for specific realizations of $Y$ and $A$, e.g., $DE_{a_i, a_j}(y|a_i)$. For calculating the bounds, we replace $P(y|m,a,z)$ through our prediction outcome for $f(m,a,z)$. Although $P(y|m,a,z)$ is a random variable (depending on $y$) for deriving standard fairness bounds when training the prediction model, it is a non-random variable, independent of $y$, i.e., $f(m,a,z)$.
> Then, following the definition of the bounds given in Thm.1, we average over $M$ and $Z$ and finally obtain a non-random quantity that only depends on the realization of $A$. We denote these quantities as
> $\mathrm{CF}\_{\mathbb{E}}^+(a_i, a_j)$ and $\mathrm{CF}\_{\mathbb{E}}^-(a_i, a_j)$ for $\mathrm{CF} \in {\mathrm{DE}, \mathrm{IE}, \mathrm{SE}}$. We employ a notation containing the expectation, to explicitly show the non-randomness and the independence of $y$.
>
> In the following, we outline further details on the derivations for each of the tasks binary classification, regression and mutli-class classification separately. We will switch from the notation used in our manuscript to the notation used in your previous question, i.e. random variables $X$,$Y$ below.
>
> First, let’s take a look at **binary variables** $X,Y$, which also corresponds to our implementation in the experiments on synthetic data. For each of the three different fairness effects, we are then only interested in, e.g., $\mathrm{CF} (Y=1 \mid X=1)$. Due to symmetry reasons for binary $X$ (e.g., discrimination of women compared to men is symmetric to discrimination of men compared to women), it is sufficient to focus on one realization of $X$.
> In our case, we thus have one constraint per $\mathrm{CF}$, i.e., three constraints and thus three Lagrange multipliers in total.
>
> Next, let us consider the **regression** setting, i.e., $Y$ continuous, $X$ binary, which corresponds to our real-world study. Here, it is reasonable to consider the expectation over $Y$. We are thus left again with three constraints in total (one per $\mathrm{CF}$ ).
>
> Finally, let’s consider **multi-class classification**. Now there are two options of defining constraints to optimize over. One option is (as you noted in your question) to treat each $\mathrm{CF}$ for each possible realization of $Y$ and thus minimize wrt. $3 \times |\mathcal{Y}|$ constraints. Although this is feasible for binary classification, it can be highly challenging for large $|\mathcal{Y}|$. Therefore, an alternative option is to constrain the expected $\mathrm{CF}$  over all realizations of $Y$. The particular choice of which notion to constrain is up to the practitioner in real-world use cases. We **added a clarification** regarding this choice in our **updated manuscript** and **updated Algorithm 1** to spell this out more explicitly.
>
> The **max function** is then applied following Eq. (11), i.e., on **each constraint separately** (=which is also what you stated in your question). Each max function only takes the maximum of the absolute value of the upper or lower bound of one specific $\mathrm{CF}$ (either averaged over all outcomes of $Y$ or one per outcome separately). Therefore, there are *# constraints*-many max functions and, thus,  *# constraints*-many Lagrange multipliers. Overall, $\lambda, c$ and $\gamma$ in the pseudo-code are **vectors** of dimension *# constraints*.
>
> The function $max(x,y)$ is differentiable unless $x=y$. In this case, one can observe that the subdifferential of $max(x,y)$ at $(x,y)$ is given by the convex hull of $[(1,0), (0,1)]$. In general, choosing any subgradient at $(x,y)$ circumvents the differentiability problems.
> To furthermore ensure computational stability, each of the gradients wrt. $x$ or $y$ is best set to 0.5, i.e., the element in $[(1,0), (0,1)]$ with the smallest norm. Common deep learning libraries directly implement this rule in the provided functions.
>
> **Action:**
> Upon reading your question, we realized that we should clarify the algorithm in our paper. To this end, we have **updated Algorithm 1** in our paper and added a **further discussion** in **Supplement E.4**. In our updated algorithm, we now state the different multipliers more explicitly. Furthermore, we have now specified the max operation in line 6 more rigorously.
>
> We are again thankful for your question, which helped us in improving our presentation. We hope that our answer helped in addressing your question satisfactorily. Please let us know if there are further things in our presentation that should be improved.

---

> > ### Comment · Reviewer_rYvN · 2023-11-22
> > **Helpful**
> >
> > This was helpful, thank you (I'd imagine the discussion would be much easier face-to-face...)
> >
> > I'd add a few things though:
> >
> > * $P(y | m, a, z)$ is not a random variable, it's the evaluation of a function on four constants $y, m, a, z$. $P(Y | m, a, z)$, $P(y | M, a, z), ..., p(Y | M, A, z), P(Y | M, A, Z)$ etc., those would be random variables as they are functions evaluated *at* random variables. (But then again I'm not sure why this is relevant...)
> >
> > * in the regression setting, if only you care about the expected value of $Y$, that's fine. But that's not what Eq. (8), (9), (10) are (they define whole functions over the space of $y$, which is confusing). Please make sure to fully spell out where these changes between whole functions and expected values take place.
> >
> > * when I mentioned $max$, I was referring to the usual mathematical programming formulation where we have $d$ constraints $g_i(x) \leq b_i$, where the intersection can be summarised as $max_{i = 1, 2, \dots, d} [g_i(x) - b_i] \leq 0$. This is far more complex than a simple $\max(x_1, x_2)$ for some unrelated pair $x_1, x_2$ of variables, there is a reason why it took a while for a practical polynomial time algorithm for linear programming to be discovered. We would typically not clump all of these constraints in one single Lagrange multiplier and differentiate through $x$.
> >
> > In any case: I can see where you are going and I think what you are doing is probably correct in your implementation although I still think presentation needs work. Please do give another pass in an eventual published version of the paper and release your code, everyone will benefit (including yourselves). I believe you have shown you are able to do that. I won't raise further questions and I will raise my score.
> >
> > Thanks again.

---

> ### Author Response · Authors · 2023-11-23
> **Response to follow-up "helpful" by reviewer rYvN**
>
> Thank you again very much for the helpful discussion on the mathematical representation of our framework. We highly value your comments and input, which have significantly improved the presentation in our paper.
>
> We will carefully revise the paper following your suggestions and make sure to fully spell out all changes between whole functions and expected values. In particular, for an eventual camera-ready version, we will revise the complete notation in our paper again to ensure consistency. If you have any additional remarks or questions about the notation, we are more than happy to incorporate them in a revised version of the paper.
>
> We also have released the code in an anonymous GitHub repository (https://anonymous.4open.science/r/FairSensitivityAnalysis_anonymous-0DA2/README.md / the link is also in the paper) to preserve anonymity. Upon acceptance, we will move our codes to a public GitHub.

---

### Author Response · Authors · 2023-11-19
**Response to all reviewers**

Thank you very much for the constructive evaluation of our paper and your helpful comments! We addressed all of them in the comments below.

Our main improvements are the following:
* We added **new theoretical results** where we derived bounds for further fairness notions (see new **Supplement F**). Thereby, we demonstrate how our framework can be applied to further fairness notions, which shows the broad applicability of our framework.
We added **new empirical results** to demonstrate the effectiveness of our framework for continuous features (see new **Supplement J**).
* We added **new empirical results** to show the robustness of our framework with respect to the sensitivity parameter used for training (see new **Supplement I.2**).
* We spell out the **novelty and importance** of our work more clearly. To this end, we added a new **Table 1**, where we show how our work contributes to different literature streams. Further, we clearly distinguish our work from existing literature on sensitivity analysis, especially the GMSM (see our revised **Section 2** and our new **Supplement C.3**).
* We added a discussion where we justify the **choice of our sensitivity model** (see new **Supplement C.2**).

We incorporated all changes into the **updated version of our paper**. Therein, we highlight all key changes in **blue color**. Given these improvements, we are confident that our paper will be a valuable contribution to the fairness as well as the causal machine learning literature and a good fit for ICLR 2024.

---

> ### Author Response · Authors · 2023-11-22
> **Response to all reviewers (update)**
>
> We want to thank reviewer rYvN for noting the ambiguity of our notation in the presentation of Algorithm 1. Based on the reviewer’s valuable suggestions, we improved the presentation in the following way:
>
> - We **improved the notation** for our constraints in Section 5 and Algorithm 1 and **added further explanations** on the calculations performed in our framework.
> - We present an **additional alternative training algorithm** (see our **new Algorithm 2**) for multi-class classification in our **new Supplement E.4** along with a discussion of both algorithms applied to different tasks, i.e., binary classification, mutli-class classification,
> and regression.
>
> We added the new changes in **red color**.

---

### Meta-Review · Area_Chair_9Sky · 2023-12-06

**Metareview:**

The paper shows how to integrate a recent framework for sensitivity analysis in the construction of a class of (causally) fair classifiers. There were some subtleties to be clarified about the novel aspects of the contribution, plus details on how the actual constrained optimization worked. Ultimately, the clarifications needed appear to be doable in a final manuscript.

It is acknowledged that the original submission exceeded the ICLR format by three lines. After consulting with the SAC, I judged that the mistake was not in bad faith and could have been easily avoided, with no discernible practical impact to the reviewing process. Hence, this oversight did not affect the recommendation of the paper.

**Justification For Why Not Higher Score:**

The methodological contribution does require quite some work to be fleshed out. The starting point, however, is a direct combination of results of a recent sensitivity analysis method and known results for identification of counterfactual path-specific effects.

**Justification For Why Not Lower Score:**

Causal modeling is one important aspect in fairness, and gaps on making the pipeline robust to causal misspecification should be filled. The manuscript provides one step in the right direction.

---

### Decision · Program_Chairs · 2024-01-16

Accept (poster)